# Understanding Fixed Predictions via Confined Regions

Connor Lawless [1]   Tsui-Wei Weng [2]   Berk Ustun [2]   Madeleine Udell [1]

## Abstract

Machine learning models can assign *fixed* predictions that preclude individuals from changing their outcome. Existing approaches to audit fixed predictions do so on a pointwise basis, which requires access to an existing dataset of individuals and may fail to anticipate fixed predictions in out-of-sample data. This work presents a new paradigm to identify fixed predictions by finding *confined regions* of the feature space in which all individuals receive fixed predictions. This paradigm enables the certification of recourse for out-of-sample data, works in settings without representative datasets, and provides interpretable descriptions of individuals with fixed predictions. We develop a fast method to discover confined regions for linear classifiers using mixed-integer quadratically constrained programming. We conduct a comprehensive empirical study of confined regions across diverse applications. Our results highlight that existing pointwise verification methods fail to anticipate future individuals with fixed predictions, while our method both identifies them and provides an interpretable description.

## 1. Introduction

Machine learning is increasingly used in high-stakes settings to decide who receives a loan (Hurley & Adebayo, 2016), a job interview (Bogen & Rieke, 2018), or an organ transplant (Murgia, 2023). Models predict outcomes using features about individuals, without considering how individuals can change them (Liu et al., 2024). Consequently, models may assign an individual a *fixed prediction* which is not *responsive* to their actions.

The responsiveness of a model is integral to its safety in settings where predictions map to people. In lending and

hiring, fixed predictions may preclude individuals from access to credit and employment. In content moderation, a fixed prediction can ensure that malicious actors are unable to evade detection by manipulating their features. There has been little work that mentions this effect, let alone studies it. Yet fixed predictions arise in practice. Recently, a predictive model used to allocate livers in the United Kingdom was found to preclude all young patients – no matter how ill – from receiving a liver transplant (Murgia, 2023).

Existing approaches to check model responsiveness (see Kothari et al., 2023) work by verifying recourse on an individual-by-individual basis. These pointwise approaches can only verify recourse for available data, and do not provide guarantees on model responsiveness out-of-sample. In practice, this means that critical issues can only be identified *after* a model has been deployed and it is too late to prevent harms. Moreover, in some settings there may be no data available to run these pointwise procedures due to privacy or because the model is being deployed on a new population. For instance, many interpretable medical and criminal justice scoring systems are available publicly (see e.g., Morrison et al., 2022; Yamga et al., 2023; Ribeiro Junior et al., 2023; of Pennsylvania, 2025), but gaining access to a representative dataset is difficult due to privacy concerns. Without available data, pointwise approaches can only be used after generating synthetic data that may not be representative of the entire population or feature space. Finally, even when individuals without recourse can be identified, it can be hard to determine the root cause of these fixed predictions (Rawal et al., 2020).

This work studies a new paradigm in algorithmic recourse that characterizes fixed predictions by verifying recourse over an *entire region* of the feature space (e.g., all plausible job applicants). In contrast to existing pointwise approaches, auditing regions of the feature space can identify *confined regions* (i.e., where all individuals are assigned a fixed prediction), or provide a formal certification of model responsiveness over the entire region. This approach is robust to distribution shifts, can be used without the need for available datasets or synthetic data generation, and provides an interpretable description of individuals with fixed predictions.

Verifying recourse for a *single individual* is a non-trivial

[1]Stanford University [2]University of California, San Diego. Correspondence to: Connor Lawless <lawlessc@stanford.edu>.

*Proceedings of the 42ⁿᵈ International Conference on Machine Learning*, Vancouver, Canada. PMLR 267, 2025. Copyright 2025 by the author(s).

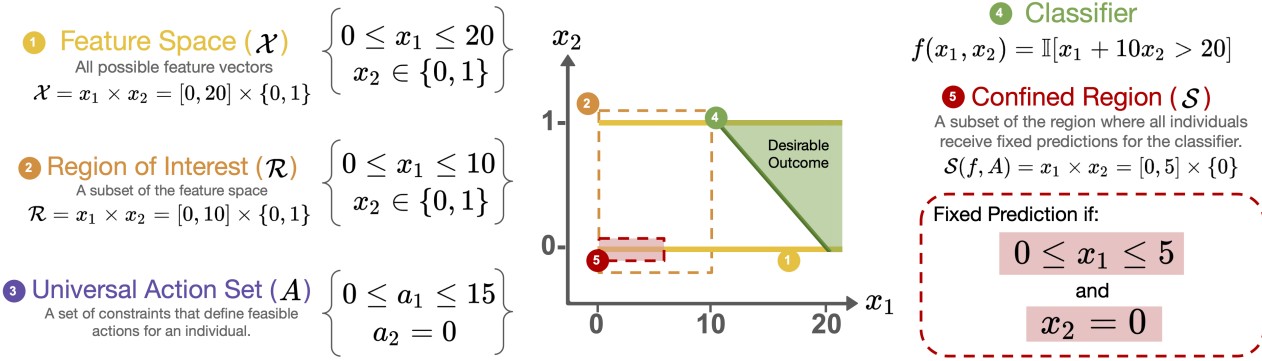

Figure 1. Stylized recourse verification problem that aims to find all individuals that are assigned a fixed prediction under a given classifier. The feature space (1) defines all possible feature vectors $\mathbf{x}$. The region of interest (2) is a subset of the feature space representing individuals on which to audit recourse. The action set (3) defines a set of constraints on what actions $\mathbf{a}$ individuals can take to change their prediction under a classifier (4). A confined region (5) is a subset of the region of interest in which all individuals are assigned a fixed prediction, which depends on both the classifier and the action set. In this example, a single confined regions provides an interpretable characterization of all individuals that receive a fixed prediction.

combinatorial problem that requires performing exhaustive search over a subset of the feature space that captures both the model as well as actionability constraints (Ustun et al., 2019; Kothari et al., 2023). Characterizing all fixed predictions represents an even more intensive setting that requires searching over any plausible individual. Despite these challenges, we develop a fast method to find confined regions by *Mixed-Integer Quadratically Constrained Programming* (MIQCP) to audit linear classifiers. The main contributions of this work include:

1. We introduce a new approach to formally verify recourse for linear classifiers over entire regions of the feature space. This tool can be used certify the responsiveness of classifiers beyond data present in a training dataset and can provide stronger guarantees for out-of-sample data. We also present tools to find (or enumerate all) confined regions in the feature space, providing an intuitive tool for model developers to audit and correct problems with model responsiveness.

2. We develop fast methods that are able to find confined regions via MIQCP. Our approach handles a broad class of actionability constraints, and can verify recourse within seconds on real-world datasets.

3. We evaluate our approach on applications in consumer finance, content moderation, and criminal justice. Our results show that pointwise verification approaches fail to verify model responsiveness over regions, emphasizing the need for tools that audit recourse beyond individual data points. We also showcase the ability of our approach to audit model responsiveness in settings with no public datasets via a case-study on the Pennsylvania criminal justice sentencing risk assessment instrument.

**Related Work**  Our work introduces a new direction for algorithmic recourse (also known as counterfactual explanations) (Venkatasubramanian & Alfano, 2020; Karimi et al., 2020b). We build on a line of work that has focused on generating *actionable* recourse under hard constraints on what actions can be made (Ustun et al., 2019; Russell, 2019; Mahajan et al., 2019; Mothilal et al., 2020; Kanamori et al., 2021; Karimi et al., 2020a). Recent work has highlighted that under *inherent* actionability constraints, ML models may assign fixed predictions that preclude access for individuals (Kothari et al., 2023; Dominguez-Olmedo et al., 2022; Karimi et al., 2021). While these works have highlighted the problem of fixed predictions, no existing work has attempted to characterize regions in which this phenomenon occurs for a given classifier.

Most of the existing work on algorithmic recourse has focused on providing or verifying recourse for individuals. These approaches fail to provide a high level understanding of recourse (i.e., what actions are needed for what types of individuals) which can be used by stakeholders to interpret and calibrate their trust in the underlying ML model. Motivated by this shortcoming, recent work has studied the problem of generating a global summary of recourse via either mapping individuals to a fixed number of possible actions (Lodi & Ramírez-Ayerbe, 2024; Ley et al., 2023; Warren et al., 2024), limiting the number of features that can be used in recourse across all instances (Carrizosa et al., 2024), or providing an interpretable summary of potential recourse options for different sub-groups (Rawal & Lakkaraju, 2020). Our work is fundamentally different from these approaches in that we aim to find and characterize confined regions (i.e., interpretable regions in the feature population without recourse) instead of global actions (i.e., interpretable regions in the action space). Our approach also

formally certifies recourse over an entire region, whereas existing approaches provide no out-of-sample guarantees.

More broadly, our work builds on a line of research that has focused on algorithmic recourse as a tool to safeguard access in applications such as hiring and lending. Towards this aim, other work has studied how to generate recourse that is robust to model updates (Upadhyay et al., 2021; Forel et al., 2024), distributional shifts (Altmeyer et al., 2023; Guo et al., 2022; Hardt et al., 2023; O'Brien & Kim, 2022; Rawal et al., 2020), imperfect adherence to the prescribed recourse (Pawelczyk et al., 2022; Virgolin & Fracaros, 2023; Maragno et al., 2024), and causal effects (Mahajan et al., 2019; Karimi et al., 2020c; Kleinberg & Raghavan, 2020).

## 2. Problem Statement

We consider a classification task of predicting a label $y \in \{0, 1\}$ from a set of $d$ features $\mathbf{x} = [x_1, x_2, \ldots, x_d] \in \mathcal{X}$ in a bounded feature set $\mathcal{X}$. We study linear classification models, a broad function class encompassing popular methods such as logistic regression, linearizable rule-based models (e.g., rule sets, decision lists), and concept-bottleneck models (Koh et al., 2020; Sun et al., 2024). We assume access to the linear classifier $f(\mathbf{x}) = \text{sign}(\mathbf{w}^\top \mathbf{x} + b)$ where $\mathbf{w} \in \mathbb{R}^d$ is the vector of coefficients and $b \in \mathbb{R}$ is the intercept of the classifier. Without loss of generality, we assume $f(\mathbf{x}) = 1$ is the desired outcome (e.g., receiving a loan). We use bold-face variables (e.g., $\mathbf{x}$) to denote vectors, and standard text with subscripts (e.g., $x_d$) to denote a specific element of a vector.

**Actionability Constraints** The recourse verification problem (Kothari et al., 2023) tests whether an individual $\mathbf{x} \in \mathbb{R}^d$ can obtain the desired outcome of a model by *actions* on their features. Each action is a vector $\mathbf{a} \in \mathbb{R}^d$ that shifts the features of the individual to $\mathbf{x} + \mathbf{a} = \mathbf{x}' \in \mathcal{X}$. We refer to the set of all actions an individual $\mathbf{x}$ can take as the *action set* $A(\mathbf{x})$. In practice, an action set is represented by a set of constraints. Table 1 shows sample deterministic actionability constraints represented in both natural language and mathematical formulae that can be embedded into an optimization problem. Actionability constraints can capture inherent limitations on how semantically meaningful features can change (e.g., age can only increase) and how those changes impact other features (e.g., increasing years of account history also increase age).

We summarize all feasible actions that lead to the desirable outcome for an individual in the recourse set.

**Definition 2.1.** The *recourse set* consists of all feasible actions for an individual $x$ that lead to the desired outcome:

$$\text{Recourse}(\mathbf{x}, f, A) = \{a : f(\mathbf{x} + \mathbf{a}) = 1, \mathbf{a} \in A(\mathbf{x})\}$$

We say an individual receives a fixed prediction under a classifier $f$ and action set $A$ if $\text{Recourse}(\mathbf{x}, f, A) = \emptyset$, and has recourse if $|\text{Recourse}(\mathbf{x}, f, A)| \geq 1$

**Verification with Regions** This paper studies the problem of verifying recourse over an entire region of the feature space $\mathcal{R} \subseteq \mathcal{X}$. This region could represent plausible characteristics of decision subjects (e.g., any loan applicants), or a sub-group of interest (e.g., all Black female loan applicants).

We start by generalizing the notions of recourse and fixed predictions to regions instead of individual data points.

**Definition 2.2.** A region $\mathcal{R}$ is *responsive* under a classifier $f$ and action set $A$ if all individuals within the region have recourse. A region $\mathcal{R}$ is *confined* if all individuals within the region have a fixed prediction.

Note that a region is responsive if *all* individuals within the region have recourse. Similarly, a region is only confined if all individuals are assigned fixed predictions. If the region contains a mix of individuals with recourse and fixed predictions, the region is neither confined nor fixed.

The goal of the *Region Recourse Verification Problem (RVP)* is to certify whether a given region $\mathcal{R}$ is confined, responsive, or neither. This task can be cast as an optimization problem that finds the largest confined area $\mathcal{S}(f, A)$ within the region $\mathcal{S} \subseteq \mathcal{R}$. For simplicity, we drop the explicit dependence of $\mathcal{S}$ on $(f, A)$. Let $\text{Size}(\mathcal{S})$ be a function that quantifies the size of the confined area $\mathcal{S}$. Given a region $\mathcal{R}$, a classifier $f$, and an action set $A$, the RVP can be modeled as the following optimization problem:

$$\begin{aligned} \underset{\mathcal{S}}{\text{maximize}} \quad & \text{Size}(\mathcal{S}) \\ \text{subject to} \quad & \forall x \in \mathcal{S} : \text{Recourse}(x, f, A) = \emptyset \\ & \mathcal{S} \subseteq \mathcal{R} \end{aligned}$$

An optimal solution to this optimization problem, $\mathcal{S}^*$, can be used to directly verify whether the entire region $\mathcal{R}$ is confined, responsive, or neither:

$$\text{Verify}(\mathcal{S}^*, \mathcal{R}) = \begin{cases} \texttt{Responsive}, & \text{if } \mathcal{S}^* = \emptyset \\ \texttt{Confined}, & \text{if } \mathcal{S}^* = \mathcal{R} \\ \bot, & \text{otherwise} \end{cases}$$

**Use Cases** Verifying recourse over regions is a powerful tool that can be used to catch potential harms that arise from fixed predictions *before* deploying a model, characterize confined regions, and audit discrimination.

*Detecting Harms before Deployment.* Existing approaches that verify recourse for individual data points may fail to find confined regions before deploying a model, especially in settings with large distribution shifts. This failure can

| Class | Example | Features | Constraint |
|---|---|---|---|
| Immutability | `n_dependents` should not change | $x_j = $ `n_dependents` | $a_j = 0$ |
| Monotonicity | `reapplicant` can only increase | $x_j = $ `reapplicant` | $a_j \geq 0$ |
| Integrality | `n_accounts` must be positive integer $\leq 10$ | $x_j = $ `n_accounts` | $a_j \in \mathbb{Z} \cap [0 - x_j, 10 - x_j]$ |
| Categorical Encoding | preserve one-hot encoding of `married`, `single` | $x_j = $ `married` 
 $x_k = $ `single` | $a_j + x_j \in \{0, 1\} \quad x_k + a_k \in \{0, 1\}$ 
 $a_j + x_j + a_k + x_k = 1$ |
| Ordinal Encoding | preserve one-hot encoding of `max_degree_BS`, `max_degree_MS` | $x_j = $ `max_degree_BS` 
 $x_k = $ `max_degree_MS` | $a_j + x_j \in \{0, 1\} \quad x_k + a_k \in \{0, 1\}$ 
 $a_j + x_j + a_k + x_k = 1 \quad a_j + x_j \geq a_k + x_k$ |
| Logical Implications | if `is_employed = TRUE` then `work_hrs_per_week` $\geq 0$ else `work_hrs_per_week` $= 0$ | $x_j = $ `is_employed` 
 $x_k = $ `work_hrs_per_week` | $a_j + x_j \in \{0, 1\}$ 
 $a_k + x_k \in [0, 168]$ 
 $a_j + x_j \leq 168(x_k + a_k)$ |
| Causal Implications | if `years_of_account_history` increases then `age` will increase commensurately | $x_j = $ `years_at_residence` 
 $x_k = $ `age` | $a_j \leq a_k$ |

*Table 1.* Examples of deterministic actionability constraints. Each constraint can be expressed in natural language and modeled using standard tools from mathematical programming (see e.g., Wolsey, 2020).

result in tangible harms such as precluding individuals from receiving loans or allowing malicious actors to bypass content filters. Verifying recourse over regions can catch these harms during model development and allow model developers to adjust the model before deployment.

*Characterizing Confined Regions.* The RVP can be run sequentially to enumerate all confined boxes in a region for a classifier. These boxes are simple to understand and can be used to help model developers debug ML models or provide a high-level summary of confined regions for stakeholders. These confined boxes can also be used to construct a valid (lower) bound on the fraction of the region that is confined. This can be used by model developers as a metric to compare two potential ML models before deployment.

*Data-Free Auditing.* Recourse verification over regions can be used as a tool to find sources of potential discrimination in a model (e.g., individuals that are assigned fixed predictions in a lending application). This approach only requires access to a classifier and a description of the region to audit (which can be as simple as bounds on each feature). This allows external auditors to evaluate the responsiveness of a classifier with *no access to the underlying data*. This is especially powerful in applications where models are publicly available but associated data is not (e.g., criminal justice (of Pennsylvania, 2025) and medicine (Yamga et al., 2023)).

**Pitfalls of Auditing by Observation** Existing methods for recourse verification (e.g., Kothari et al., 2023) can only verify recourse for observed data (e.g., individuals in a training dataset). These point-wise approaches can be used to audit the responsiveness of a region by verifying whether any observed data points are assigned a fixed predictions. However, these approaches may fail to correctly output whether a region is responsive or confined. We outline two kinds of failures:

**Definition 2.3.** Given a recourse verification task for a region $\mathcal{R}$, a model $f$, and action set $A$, we say that a method returns a *blindspot* if it outputs that a region is responsive

but there exists an individual in the region that is assigned a fixed prediction.

**Definition 2.4.** Given a recourse verification task for a region $\mathcal{R}$, a model $f$, and action set $A$, we say that a method returns a *loophole* if it outputs that a region is confined but there exists an individual in the region with recourse.

These failure modes can arise when an audit over observed data does not reveal all potential individuals within the region. In Section 4 we show that this can occur with a variety of different strategies to select data to test within a region.

## 3. Verification via Confined Boxes

Towards formally verifying recourse over an entire region, we formulate a *mixed-integer quadratically constrained program* (MIQCP) to solve the RVP.

**Characterizing Regions with Boxes** We focus on a special case of the RVP that finds the largest confined *box*. A box is a set defined by simple upper and lower bound constraints on each dimension. Let $U_j = \max_{x \in \mathcal{R}} x_j$, $L_j = \min_{x \in \mathcal{R}} x_j$ be the upper and lower bound for each feature $j$ in the region. Given an upper bound, $\mathbf{u} \in \mathbb{R}^d : \mathbf{u} \leq \mathbf{U}$, and lower bound, $\mathbf{l} \in \mathbb{R}^d : \mathbf{l} \geq \mathbf{L}$, a box $B_{\mathcal{R}}(\mathbf{u}, \mathbf{l})$ is defined as $B_{\mathcal{R}}(\mathbf{u}, \mathbf{l}) = \{\mathbf{x} \in \mathcal{R} : \mathbf{l} \leq x \leq \mathbf{u}\}$. We focus on boxes due their interpretability, which can help model developers understand the source of fixed predictions. Boxes can be viewed as a type of *decision rule*, which have been widely studied for their interpretability within the broader ML community (e.g., (Lawless et al., 2023; 2022; Lawless & Gunluk, 2023)). For ease of notation we drop the explicit dependence on $\mathcal{R}$ and refer to boxes as $B(\mathbf{u}, \mathbf{l})$. We define the size of a box $B(\mathbf{u}, \mathbf{l})$ as the sum of the normalized ranges of each feature:

$$\text{Size}(B(\mathbf{u}, \mathbf{l})) = \sum_{j=1}^{d} \frac{u_j - l_j}{U_j - L_j} \qquad (1)$$

**Generating Confined Boxes** We start by formulating the related problem of auditing whether a given box $B(\mathbf{u}, \mathbf{l})$ in region $\mathcal{R}$ contains any data points with recourse, which we denote the *Region Recourse Existence Problem (REP)*. Let $\mathbf{x} \in \mathbb{R}^{d-q} \times \mathbb{Z}^q$ be a decision variable representing an individual, and $\mathbf{a} \in \mathbb{R}^{d-q} \times \mathbb{Z}^q$ represent an action. We assume that the region $\mathcal{R}$, feature space $\mathcal{X}$, and action set $\mathcal{A}$ can be represented by a set of constraints over a mixed-integer set (see Figure 1 for an example). This general assumption encompasses a variety of potential regions and feature sets. We model the REP as a mixed-integer linear program (MILP) over $\mathbf{x}$ and $\mathbf{a}$ (see Appendix A for details).

Recall that the RVP can be cast as an optimization problem to find the largest confined region within $\mathcal{R}$. By definition the REP is infeasible for *every confined box*. To certify that the REP is infeasible for a given box, and by extension certify that the box is confined, we leverage a classical result from linear optimization called Farkas' lemma:

**Theorem 3.1** (Farkas (1902))**.** Let $A \in \mathbb{R}^{m \times n}$ and $b \in \mathbb{R}^m$. Then exactly one of the following two assertions is true:

    I. There exists $x \in \mathbb{R}^n$ such that $Ax \leq b$

    II. There exists $y \geq 0$ such that $A^T y = 0$ and $b^\top y = -1$

Farkas' lemma states that we can certify that a system of inequalities over continuous variables $Ax \leq b$ is infeasible by finding a *Farkas certificate* $y \geq 0$ such that $A^\top y = 0$ and $b^\top y = -1$. In our context, we can thus view the problem of finding a confined box as a joint problem of selecting a box and finding an associated Farkas certificate for the REP. However, Farkas' lemma only applies to *continuous* variables, and the REP can include discrete variables.

We extend Farkas' certificates to the discrete setting using a simple strategy that simultaneously generates certificates for all possible continuous restrictions of the REP. A *continuous restriction* of a MILP is a restricted version of the optimization problem where all discrete variables are fixed to specific values. Note that a box is confined if and only if every continuous restriction of the REP is infeasible.

Let $\mathcal{C}$ be the set of continuous restrictions, where each restriction $c \in \mathcal{C}$ corresponds to a specific set of fixed values for the discrete variables (e.g., $x_1 = 1, x_2 = 2$ for a problem with two discrete variables $x_1, x_2 \in \mathbb{Z}^2$). Note that the set $\mathcal{C}$ is finite, from the assumption $\mathcal{R}$ is bounded and only discrete variables are fixed, but grows exponentially with respect to the number of discrete variables. If there are no discrete variables in the REP, there is a single continuous restriction representing the full problem with no fixed values. In settings where there are a large number of discrete variables, enumerating all possible continuous restrictions may become computationally intractable. However, we prove in Section 3.1 that under very general constraints and minimal

assumptions we can relax many if not all of the discrete variables in the REP. Under these new theoretical results, the set of restrictions that the algorithm must consider is often incredibly small (e.g., $|\mathcal{C}| \leq 4$ for all the datasets and actionability constraints considered in Kothari et al. (2023)).

We formulate a continuous restriction $c \in \mathcal{C}$ of the REP as a linear program (LP) (see Appendix A), which we represent in the following standard form:

$$C_c \mathbf{x} + D_c \mathbf{a} \leq b_c(\mathbf{u}, \mathbf{l})$$

where $C_c$ and $D_c$ are $m \times d$ matrices and $b_c(u, l)$ is a $m$-dimensional vector that is a linear function of the box upper and lower bounds $\mathbf{u}, \mathbf{l}$. Here $m$ represents the number of constraints in the continuous restriction of the REP.

**MIQCP Formulation** We can now formulate the RVP as MIQCP that finds the largest box with Farkas certificates of infeasibility for every continuous restriction. Let $\mathbf{y}_c \in \mathbb{R}^m$ be decision variables representing the Farkas certificate for a continuous restriction $c \in \mathcal{C}$, and $\mathbf{u}, \mathbf{l} \in \mathbb{Z}^d$ represent the upper and lower bounds of a box. Note that there is one variable in $\mathbf{y}$ for every constraint in the continuous restriction. We can now find the largest confined box $B(\mathbf{u}, \mathbf{l})$ with associated certificates of infeasibility $y_c$ for $c \in \mathcal{C}$ using the *Farkas Certificate Problem (FCP)*:

$$
\begin{align}
\underset{\mathbf{y}_c, \mathbf{u}, \mathbf{l}}{\text{maximize}} \quad & \sum_d \frac{u_d - l_d}{U_d - L_d} \tag{2a} \\
\text{subject to} \quad & b_c(\mathbf{u}, \mathbf{l})^\top \mathbf{y}_c = -1 && \forall c \in \mathcal{C} \tag{2b} \\
& C_c^\top \mathbf{y}_c = 0 && \forall c \in \mathcal{C} \tag{2c} \\
& D_c^\top \mathbf{y}_c = 0 && \forall c \in \mathcal{C} \tag{2d} \\
& \mathbf{y}_c \geq 0 && \forall c \in \mathcal{C} \tag{2e} \\
& \mathbf{L} \leq \mathbf{l} \leq \mathbf{u} \leq \mathbf{U} \tag{2f} \\
& \mathbf{u}, \mathbf{l} \in \mathbb{Z}^d \tag{2g}
\end{align}
$$

The objective of the problem is to maximize the size of the box, as defined in Equation (1). Constraints (2b)-(2e) follow from Farkas' lemma and ensure that $y_c$ is a valid certificate of infeasibility for the continuous restriction $c$ of the REP. Constraint (2f) ensures the FCP generates a valid box within the region $\mathcal{R}$. We restrict $\mathbf{u}, \mathbf{l}$ to be integer variables to prevent numerical precision issues when solving this MIQCP in practice. This is not an onerous assumption as any continuous variable $x_j$ with a desired precision $10^{-p}$ can be re-scaled and rounded to an integer variable $10^p x_j$. The problem is quadratically constrained due to the inner product of $b_c(\mathbf{u}, \mathbf{l})$ and $\mathbf{y}_c$ in constraint (2b). While MIQCPs are often more computationally demanding than MILPs, the FCP can be solved in seconds on real-world datasets using commercial solvers (e.g., Achterberg, 2019), as the problem scales with the number of features and actionability con-

straints (which are typically small) rather than the number of data points in the data set.

When verifying recourse over a *fixed* box $B(\mathbf{u}, \mathbf{l})$ the FCP can be decomposed into $|\mathcal{C}|$ problems (solved independently for each continuous restriction). If the FCP is infeasible for any continuous restriction $c$, then the RVP is infeasible for the box. If the FCP is feasible for all continuous restrictions $c \in \mathcal{C}$, then the box is responsive. Alas, when optimizing over potential boxes, the FCP cannot be decomposed as the variables $\mathbf{u}, \mathbf{l}$ link all the continuous restrictions.

**Generating Multiple Boxes**   Solving an instance of the FCP generates a single confined box or certifies that the region is responsive. However, in practice, a given region may contain multiple confined regions. To provide model developers and stakeholders with a comprehensive view of individuals with fixed predictions, the FCP can be run sequentially to enumerate multiple (or all) confined boxes with the region. It does so by iteratively adding *no-good cuts* to exclude previously discovered confined regions from $\mathcal{R}$ (see Appendix B for details).

### 3.1. Handling Discrete Variables

In the preceding section, the RVP was solved by enumerating and finding Farkas' certificates for all continuous restrictions of the REP. However, this approach scales exponentially with respect to the number of discrete variables in the REP. In this section, we show that under a very broad set of actionability constraints and general assumptions we can relax all the discrete variables in the REP and still verify recourse over the entire region.

**Linear Recourse Constraints**   We consider a restricted set of constraints, which we call *linear recourse constraints* (detailed in Table 2). These constraints include a broad class of actionability constraints such as monotonicity, categorical encodings, and immutability. They can be used to define the feature space $\mathcal{X}$, the region $\mathcal{R}$, or the action set $A$. Linear recourse constraints encompass many actionability constraints considered in previous literature including all the constraints in (Ustun et al., 2019; Russell, 2019; Kothari et al., 2023). We denote an action set comprised only of these constraints as *linear recourse constraints*.

**Key Result**   Theorem 3.2 shows that we can recover the solution to the REP by solving a *linear relaxation* if:

A.1 No variable appears in more than one $K$-hot constraint.

A.2 The directional linkage constraints do not enforce relationships between variables appearing in $K$-hot constraints.

A.3 The directional linkage constraints do not imply any

circular relationships between variables.

Practically, Theorem 3.2 shows we can solve the FCP with a single continuous restriction (i.e., $|\mathcal{C}| = 1$), relaxing all discrete variables in the problem.

**Theorem 3.2.** *Under Assumptions A.1- A.3, the linear relaxation of the REP is feasible iff the REP is feasible for any problem with linear recourse constraints.*

For a full proof and formal definitions of the assumptions, see Appendix C. The assumptions for Theorem 3.2 are general and hold in many realistic settings. For instance, $K$-hot constraints are often used to encode categorical features (e.g., via a one-hot encoding). Assumption A.1 holds in this setting as each associated variable only corresponds to one encoding (i.e., one $K$-hot constraint). Similarly, Assumption A.2 holds as long as there are no logical implications between the categorical features. Finally, Assumption A.3 holds as long as there are no circular implications between variables. Circular implications between variables represent flaws in constructing the action set and should be caught prior to solving the RVP.

Theorem 3.2 holds under linear recourse constraints but not under more general constraints. In Appendix D we discuss how to extend our approach to general constraints, and provide practical guidelines on how to select continuous restrictions to include in the FCP.

## 4. Experiments

We present experiments showing that point-wise approaches fail to correctly find confined regions, while our methods to audit recourse over regions accurately find such regions when they exist. Hence our methods can help decision-makers avoid harms from deploying models with fixed predictions. We include code to reproduce our results at `https://github.com/conlaw/confined_regions/` and provide additional details and results in Appendix E.

**Setup**   We evaluate our approach on three real-world datasets in consumer finance (`heloc` (FICO, 2018), `givemecredit`(Kaggle, 2011)) and content moderation (`twitterbot` (Gilani et al., 2016)). Each dataset include features that admit *inherent* actionability constraints that apply to all individuals (e.g., preserving feature encoding) which we use to construct the action set (see Appendix E for details). We encode all categorical features using a one-hot encoding and discretize all continuous features. We split the processed dataset into a training sample (50% used to train the model), and an audit sample (used to evaluate responsiveness in deployment).

We use the training dataset to fit a $\ell_1$-regularized logistic re-

| Class | Description | Formulation | Discussion |
|---|---|---|---|
| Integer Bound Constraints | Places an integer upper or lower bound on a variable | $L_j \leq v_j \leq U_j$. | Encompasses a wide range of separable constraints including monotonicity, actionability, and bounds on the action step size (Kothari et al., 2023) |
| $K$-Hot Constraint | Preserves that the unweighted sum of a set of variables $\{v_j\}_{j \in J}$ is at most $K \in \mathbb{Z}$. | $\sum_{j \in J} \pm v_j \leq K$. | Generalizes the popular *one-hot encoding* for categorical variables. |
| Directional Linkage Constraints | Ensures that one feature, $v_j$ is greater than or equal to another feature $v_k$ | $v_j \leq v_k$. | Ensures a broad class of non-separable constraints (i.e., constraints that act on multiple features) including thermometer encodings, and deterministic causal constraints (e.g., increasing years of account history implies a commensurate increase in Age). |

*Table 2.* Linear Recourse Constraints Classes. Variables $v_j$ used in the constraints may represent $x$ variables (i.e., constrain the region), $a$ variables (i.e., constrain the actions), or $x + a$ (i.e., constrain the resulting feature vector). This restricted set of constraints encompasses a broad set of existing actionability constraints considered in previous literature.

gression model and tune its parameters via cross-validation. We use the auditing dataset to verify recourse over a set of regions $\Omega$ that represent different sub-groups of interest for the classification task. We generate these regions $\mathcal{R} \in \Omega$ by restricting the feature space to fixed combinations of immutable characteristics (e.g., all individuals with a specified age and gender). We remove all regions from $\Omega$ that do not have at least 5 data points in the training dataset. The fraction of data points with fixed predictions ($p$) in the each dataset varies between $10 - 55\%$. We represent the feature space $\mathcal{X}$ for each dataset as the smallest box containing all available data.

**Methods** We compare our method to pointwise baselines that audit recourse over a sample of individual data points to generate outputs for the entire region. Given a sample of individual data points, these point-wise approaches output that a region is responsive (confined) if all data points have recourse (no recourse). We generate different baselines by using different strategies to select which individual data points to include in the sample.

- Data: We use all data points from the training dataset that fall within the region.

- Region: We sample 100 data points uniformly at random from the region being audited.

- Score: We evaluate the data points within the region with the highest and lowest classifier score (i.e. $\text{argmax}_x w^T x$ and $\text{argmin}_x w^T x$).

- ReVer: We implement our approach, which we call **Re**gion **Ver**ification (ReVer), in Python using Gurobi (Achterberg, 2019) to solve all MIQCPs.

**Results** We summarize our results in Table 3 for all datasets and methods. We use ReVer to certify whether each region is responsive, confined, or neither. We use these results to evaluate the reliability of baseline methods. We evaluate each method in terms of the percentage of regions where it: *certifies responsive*, *outputs responsive*, outputs a *blindspot* (i.e., misses individuals in the region with fixed

| | | PointWise | | | |
|---|---|---|---|---|---|
| **Dataset** | **Metrics** | Data | Region | Score | ReVer |
| | Certifies Responsive | — | — | — | 54.2% |
| `heloc` | Outputs Responsive | 91.6% | 66.5% | 71.0% | 54.2% |
| $n = 5842$  $d = 43$ | ↳ Blindspot | 37.4% | 12.3% | 16.8% | **0.0%** |
| $\|\Omega\| = 155$  $p = 22.2\%$ | Certifies Confined | — | — | — | 0.0% |
| FICO (2018) | Outputs Confined | 0.6% | 0.0% | 0.0% | 0.0% |
| | ↳ Loophole | 0.6% | **0.0%** | **0.0%** | **0.0%** |
| | Certifies Responsive | — | — | — | 60.1% |
| `givemecredit` | Outputs Responsive | 72.2% | 60.1% | 62.9% | 60.1% |
| $n = 120,268$  $d = 23$ | ↳ Blindspot | 12.0% | **0.0%** | 2.8% | **0.0%** |
| $\|\Omega\| = 715$  $p = 7.4\%$ | Certifies Confined | — | — | — | 18.3% |
| Kaggle (2011) | Outputs Confined | 19.4% | 19.2% | 19.2% | 18.3% |
| | ↳ Loophole | 1.1% | 0.8% | 0.8% | **0.0%** |
| | Certifies Responsive | — | — | — | 25.0% |
| `twitterbot` | Outputs Responsive | 40.0% | 25.0% | 25.0% | 25.0% |
| $n = 1438$  $d = 21$ | ↳ Blindspot | 15.0% | **0.0%** | **0.0%** | **0.0%** |
| $\|\Omega\| = 20$  $p = 55.3\%$ | Certifies Confined | — | — | — | 5.0% |
| Gilani et al. (2016) | Outputs Confined | 25.0% | 25.0% | 25.0% | 5.0% |
| | ↳ Loophole | 20.0% | 20.0% | 20.0% | **0.0%** |

*Table 3.* Overview of results for all datasets, regions, and methods. For each dataset, we include the number of regions we audit ($|\Omega|$), and the fraction of data points with fixed predictions ($p$).

predictions), *certifies confined*, *outputs confined*, and outputs a *loophole* (i.e., misses individuals in the region with recourse). Additional metrics are reported in Appendix E.

**On Preempting Harms during Model Development** Our results demonstrate that individualized recourse verification approaches fail to properly predict whether a region is responsive or confined. All baseline approaches result in blindspots, ranging from 2.8% to 37.4% of regions, and loopholes, ranging from 0.6% to 20% of regions. The baseline approaches incorporate both separable and non-separable actionability constraints. Consequently, these blindspots and loopholes arise from focusing on individual data points instead of the region as a whole and highlight the importance of region-specific methods. In Appendix F we show that these failures are exacerbated in settings whether there is a distribution shift in the test dataset.

Blindspots and loopholes represent failure cases that undermine the benefits of algorithmic recourse and lead to tangible harms if left undetected in model development.

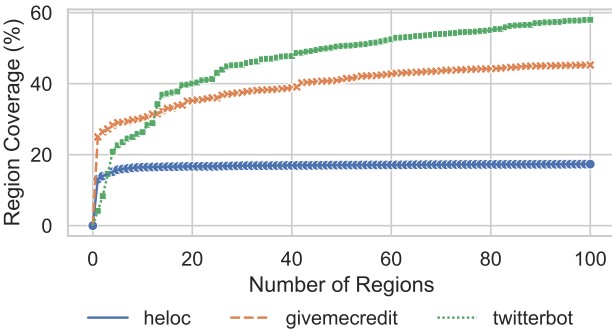

*Figure 2.* Percentage of feasible points in the region with a fixed prediction as a function of the number of confined boxes generated. Note that any point is a valid lower bound on the percentage of points with fixed predictions across the entire region.

Consider a content moderation setting where a machine learning model is used to remove sensitive content. A loophole could represent malicious content (e.g., discriminatory or offensive content) that is able to bypass the filter by changing superficial features of the post. In a consumer finance application, a blindspot represents unanticipated individuals that receive fixed predictions and are precluded from ever getting access to a loan. In both these settings, rectifying the problem after deployment would involve training a new machine learning model (e.g., by dropping features that lead to fixed predictions, or adding new features that promote actionability) which can be time-consuming, costly, and continues to inflict harm while the existing model operates. This highlights the importance of auditing recourse over regions — it foresees potential harms that occur when deployed models assign fixed predictions.

**On Characterizing Fixed Predictions**  In Section 5 we show sample regions that our algorithm certifies as confined. These regions are represented by simple decision rules (e.g., `Age 21-25` and `Male`) and can help model developers debug the sources of fixed predictions. For instance, the latter example may prompt a model developer to remove features related to age and gender from the dataset.

In some settings, fixed predictions are rare and can be represented by a small number of regions (see e.g., the case study in Section 5). However, in complicated settings there may be a large a number of confined regions. To demonstrate this phenomenon, we run our approach sequentially to enumerate up to 100 confined regions within our benchmark datasets. For each dataset, we audit a region encompassing any plausible individual (i.e., any individual satisfying indisputable conditions on each feature). Figure 2 shows the fraction of the entire region covered after generating up to 100 confined regions using our approach. Note that any point on this curve represents a valid lower bound on the total fraction of the region that is assigned a fixed prediction.

These lower bounds can be used by model developers to decide between alternative ML models for a given application. More broadly, our results highlight the scale and difficulty of fully characterizing confined regions. Although each dataset has fewer than 50 features, at least 100 regions are confined. Our results highlight that fixed predictions arise in complex ways from immutable features. As such, they create an insidious new kind of discrimination: unlike traditional forms of discrimination based on protected characteristics (e.g., race and gender), this form of discrimination is much harder to identify, requires looking at combinations of features, and depends on the classifier.

**On Computation**  Remarkably, our approach runs in *under 5 seconds* on average across all datasets (see Appendix E). This shows that although region verification is more computationally challenging than individual verification, our approach can verify regions extremely quickly. On two thirds of the datasets, our approach is even faster than the Region baseline which audits 100 individual data points.

## 5. Demonstration

**Setup**  We showcase the potential of region verification as a tool for auditing discrimination via a case study of the Pennsylvania Criminal Justice Sentencing Risk Assessment Instrument (SRAI) (of Pennsylvania, 2025). The SRAI is a simple linear scoring system that uses 37 features, including age and gender, to predict the risk of an offender committing a re-offense (i.e., criminal recidivism). The guidelines state that an offender is deemed low-risk if the SRAI risk score is under 5 points.

We audit whether there are any protected groups that are precluded from being predicted as low-risk. To that end, we specify an action set that does not allow changes to protected characteristics (i.e., age and gender), and enforces logical constraints on the features (e.g., to have a prior conviction for violent crime the number of previous convictions needs to be at least one). Full details of the actionability constraints are included in Appendix G. Note that we allow every *unprotected* characteristic (i.e., criminal history) to be mutable. This choice allows us to answer whether any protected group, based solely on protected characteristics, is never predicted to be low risk.

**Results**  We find two confined regions:

Our results highlight that the SRAI discriminates against individuals on the basis of gender and age by precluding those groups from ever receiving a desirable outcome. Our tool is able to find these forms of discrimination and present an interpretable summary of the results to stakeholders, who must then gauge whether this form of discrimination should be allowed in the criminal justice context. Notably, our

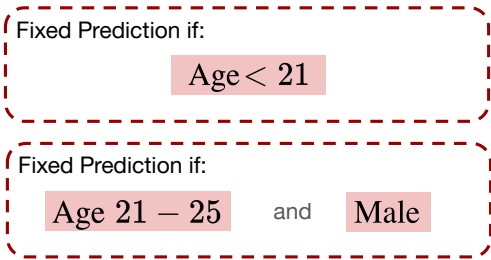

audit required no access to the data that was used to develop the SRAI, underscoring that auditing classifiers for confined regions is possible even when models are publicly available but associated data is not, as in contexts like criminal justice and medicine.

## 6. Concluding Remarks

Our paper introduces a new paradigm for algorithmic recourse that seeks to characterize fixed predictions by finding confined regions, areas in the feature space where a model is not responsive to individuals' actions. This work highlights that characterizing confined regions can help model developers pre-empt harms that arise from deploying models with fixed predictions. Our work develops a method to tackle this problem by leveraging tools from MIQCP to find confined *boxes* within a region of the feature space. Our method provides interpretable descriptions of confined regions, can be run in seconds for real-world datasets, and enables data-free auditing of model responsiveness. However, these methods should be extended to address the following limitations:

- Our methods are designed to work with linear classifiers. In principle, our methods can be extended to any model that can be represented by a MILP. However, many MILP-representable models (e.g., tree ensembles) would require solving a large, computationally-intractable MIQCP and need new algorithmic approaches.

- Our approach finds confined *boxes*. Boxes are an interpretable way to characterize regions but have limited expressive power. Future work should explore whether more expressive classes capture confined regions with fewer items.

## Impact Statement

This paper presents work whose goal is to advance the field of Machine Learning. There are many potential societal consequences of our work, none which we feel must be specifically highlighted here.

## Acknowledgements

MU and CL gratefully acknowledge support from the National Science Foundation (NSF) Award IIS-2233762, the Office of Naval Research (ONR) Awards N000142212825, N000142412306, and N000142312203, IBM, and the Alfred P. Sloan Foundation. BU and LW gratefully acknowledge support from the National Science Foundation (NSF) under award IIS-2313105.

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

# Appendix
# Understanding Fixed Predictions via Confined Regions

## A. REP Integer Programming Formulation

Recall the following:

- $\mathbf{x} \in \mathbb{R}^{d-q} \times \mathbb{Z}^q$ is a decision variable representing an individual.

- $\mathbf{a} \in \mathbb{R}^{d-q} \times \mathbb{Z}^q$ is a decision variable representing an action.

- $f(\mathbf{x}) = \text{sign}(\mathbf{w}^\top x + b)$ is the linear classifier where $f(\mathbf{x}) = 1$ is the desirable outcome.

- $\mathcal{X}$ represents the feature space.

- $\mathcal{R}$ represents the region of interest.

- $A(\mathbf{x})$ represents a set of feasible actions for a given individual $\mathbf{x}$ under the set of actionability constraints.

We assume that $\mathcal{X}$, $\mathcal{R}$, and $A$ can all be represented via a set of constraints in a MILP optimization model. We formulate the REP for a given region of interest $\mathcal{R}$ as the following MILP:

$$\mathbf{w}^\top(\mathbf{x} + \mathbf{a}) \geq b \tag{3a}$$
$$\mathbf{x} + \mathbf{a} \in \mathcal{X} \tag{3b}$$
$$\mathbf{x} \in \mathcal{R} \tag{3c}$$
$$\mathbf{a} \in A(x) \tag{3d}$$
$$\mathbf{x} \in B(\mathbf{u}, \mathbf{l}) \tag{3e}$$
$$\mathbf{x}, \mathbf{a} \in \mathbb{R}^{d-m} \times \mathbb{Z}^m \tag{3f}$$

Note that this problem has no objective as it is a *feasibility* problem. Constraint (3a) ensures that the data point $\mathbf{x}$, after taking action $\mathbf{a}$, is classified with the desirable outcome. Constraints (3c) and (3b) ensure that $\mathbf{x} + \mathbf{a}$ if a valid feature vector (i.e., included in $\mathcal{X}$), and $\mathbf{x}$ is part of the region of interest. Constraint (3d) ensures that the action $\mathbf{a}$ satisfies the actionability constraints. Finally Constraint (3e) ensures that $x$ is included in the box $B(u, l)$. Note that any solution to the REP represents a feasible $x \in B(u, l)$ with recourse. Thus the region $B(u, l)$ is *confined* if and only if the REP is infeasible.

### A.1. Continuous Restriction

Recall that $\mathcal{C}$ is the set of *continuous restrictions* of the REP, where a continuous restriction $c \in \mathcal{C}$ corresponds to the REP with fixed values for the discrete variables (e.g., $x_1 = 1, x_2 = 2$ for discrete variables $\mathcal{J}_D = \{x_1, x_2\}$). Let $v_j \in \mathcal{J}_D$ represent a discrete variable, and $s_j$ be its fixed value in continuous restriction $c$. We can incorporate continuous restrictions into the MILP formulation by adding the following additional constraints that fix the values of discrete variables in the formulation:

$$v_j = s_j \;\; \forall v_j \in \mathcal{J}_D$$

Note that since all discrete values are fixed, this is not a *linear* program over the $d - q$ continuous variables.

## B. Generating multiple confined boxes

Solving an instance of the RVP can generate *single* confined box. However in practice, a given region may contain multiple confined boxes. To provide model developers or stakeholders with a holistic view of fixed predictions, the RVP can be run sequentially to enumerate multiple (or all) confined boxes with the region. We do so by adding *no-good cuts* to the FCP that exclude previously discovered regions. This is equivalent to re-solving the RVP for a new region $\mathcal{R}' \subset \mathcal{R}$ that excludes existing confined boxes.

Let $\bar{\mathbf{u}}, \bar{\mathbf{l}}$ be an existing confined box. Let decision variables $\mathbf{z}_u, \mathbf{z}_l \in \{0,1\}^d$ track whether the new box $\mathbf{u}, \mathbf{l}$ is outside the existing box. We model the no-good cuts that exclude $\bar{\mathbf{u}}, \bar{\mathbf{l}}$ via the following linear constraints:

$$u_d \leq \bar{l}_d - 1 + (U_d - \bar{l}_d - 1)(1 - z_{ud}) \quad \forall d \in [d] \tag{4a}$$

$$l_d \geq \bar{u}_d + 1 + (L_d - \bar{u}_d + 1)(1 - z_{ld}) \quad \forall d \in [d] \tag{4b}$$

$$\sum_d z_{ud} + z_{ld} \geq 1 \tag{4c}$$

$$\mathbf{z}_u, \mathbf{z}_l \in \{0,1\}^d \tag{4d}$$

Constraint (4a) checks whether the upper bound for feature $d$ in the new box is less than the lower bound for feature $d$ in the existing box. Note that these constraints are enforced if $z_{ud} = 1$. Constraint (4b) checks an analogous condition for the lower bound of feature $d$. Constraints (4c)-(4d) ensure that at least one constraint of type (4a) or (4b) are active. In other words, it ensures that the new region must fall outside the previous region by ensuring that at least one feature differs in the new box. We exclude constraints for bounds that are tight with the population bounds (i.e., $u_d = U_d$ or $l_d = L_d$) as they are implied in the FCP. Given that our procedure tends to generate sparse regions (i.e., regions that change few features from their population upper and lower bounds), these no good-cuts typically add a very small number of binary variables and constraints to the full formulation.

# C. Proof of Theorem 3.2

We prove this result by showing that the polyhedron defining feasible individuals $\mathbf{x}$ and actions $\mathbf{a}$ under linear recourse constraints is *totally unimodular*, which means that all extreme points of the polyhedron are integral. Consequently, the linear relaxation of the REP is feasible if and only the discrete REP is feasible.

We start by (re)-introducing important notation:

- $\mathbf{x}$ is a decision variable corresponding to an individual with the region $\mathcal{R}$.

- $\mathbf{a}$ is a decision variable corresponding to an action which must be *feasible* under the action set $A(x)$.

- $v_j$ represents a set of variables corresponding to feature $j$ (i.e., $x_j, a_j$, or $x_j + a_j$).

Consider the following mixed-integer polyhedron that represents all feature space, region, and actionability constraints.

$$\mathbf{x} + \mathbf{a} \in \mathcal{X} \tag{5a}$$
$$\mathbf{x} \in \mathcal{R} \tag{5b}$$
$$\mathbf{x} \in B(\mathbf{u}, \mathbf{l}) \tag{5c}$$
$$\mathbf{a} \in A(x) \tag{5d}$$
$$\mathbf{x}, \mathbf{a} \in \mathbb{R}^{d-m} \times \mathbb{Z}^m \tag{5e}$$

Recall that for Theorem 3.2, we consider a subset of possible constraints used in $\mathcal{X}, \mathcal{R}, A$ called *linear recourse* constraints which were introduced in Section 3.1. We denote the polyhedron 5 with only linear recourse constraints as the *linear recourse polyhedron*. We repeat these constraint types here to keep this section self-contained:

- $K$-**Hot Constraints:** Let $J_i$ be the set of variables that participate in a K-hot constraint $i$. A K-Hot constraint is:

$$\sum_{j \in J_i} \pm v_j \leq K.$$

- **Unit Directional Linkage Constraints:** This constraint acts on two sets of variables $v_i$, $v_k$ and requires:

$$v_j \leq v_k.$$

- **Integer Bound Constraints:** Act on a single variable $v_j$ and require:

$$L_j \leq v_j \quad \text{or} \quad v_j \geq U_j$$

Note that in all these constraints, all variable coefficients in these constraints are in $\{0, \pm 1\}$, and thus any matrix comprised of linear recourse constraints is a $\{0, \pm 1\}$-matrix. Let $\mathcal{J} = \{J_i\}$ be the set of $K$-hot constraints. We define an undirected graph which captures all directional implications between variable indices, which we call the *implication graph*. For every variable index $i$ we create one node in the graph $n_i$ in the implication graph. We create an edge from node $n_j$ to $n_k$ if there exists a unit-directional linkage that acts on $j$ and $k$. Note that this implication graph is could include multiple connected (and potentially cyclic) components.

We now formally define assumptions A.1-A.3:

A.1 No variable appears in more than one $K$-hot constraint. Formally,

$$|J_i \cap J_k| = \emptyset, \quad \forall J_i, J_k \in \mathcal{J}.$$

A.2 The directional linkage constraints do not enforce relationships between variables appearing in $K$-hot constraints. Formally, each connected component in the implication graph contains at most one node $n_i$ corresponding to a variable in $\cup_{J_i \in \mathcal{J}} J_i$.

A.3 The directional linkage constraints do not imply any bi-directional implications between variables. Formally, the implication graph is acyclic.

We start by proving the key lemma for our result which states the linear recourse polyhedron is totally unimodular.

**Lemma C.1.** *The linear recourse polyhedron is totally unimodular.*

*Proof.* We prove this result by showing that every column submatrix of the linear recourse polyhedron admits an equitable bicoloring (Conforti et al., 2014). We consider a linear recourse polyhedron comprised exclusively of constraints where $v_j = x_j + a_j$, which we denote with the matrix Z. Note that any constraint where $v_j = x_j$ or $v_j = a_j$ uses a subset of the variables $\{x_j, a_j\}$ and thus represents a column sub-matrix of this general case. In other words, proving our result in this special case also proves the result for settings where constraints may have $v_j = x_j$ or $v_j = a_j$.

Consider the following coloring scheme for any column submatrix of Z.

1. For each index $j$, if columns corresponding to both $x_j$ and $a_j$ are included in the submatrix, color the column corresponding to $x_j$ red and the column corresponding to $a_j$ blue.

2. For each $K$-hot constraint $i$, let $\bar{J}_i \subseteq J_i$ be the remaining variables in the constraint, whose columns are included in the submatrix but have not yet been colored. Alternate coloring variables in $\bar{J}_i$ red and blue.

3. Consider the implication graph created by the directional constraints. Remove any node corresponding to an index $j$ that contains no variables selected in the submatrix, and any index $j$ where all variables corresponding to the index that are present in the submatrix have already been colored in Step 1. Note that every node $n_j$ now corresponds to exactly one column (i.e., $x_j$ or $a_j$). For each connected component of the revised implication graph, pick an initial node as follows. If there is a node $n_j$ corresponding to a variable in $\cup_{J_i \in \mathcal{J}} J_i$ in the component, select it as the initial node. Note that $n_j$ must have already been assigned a color in Step 2. If no such node exists, select an arbitrary node as the initial node. If the initial node corresponds corresponds to a column that has not yet been colored, color the corresponding column arbitrarily. We now traverse the connected component coloring columns as follows. Given a current node $n_j$ with color $c$, color all nodes $n_k$ connected to it that are uncolored with the opposite color. Repeat until all nodes in the connected component are colored.

4. Note that any remaining uncolored columns must correspond to a single variable $x_j$ or $a_j$ that only participate in integer bound constraints. Color each column arbitrarily.

Note that assumptions A.1- A.3 ensure that this is a valid coloring (i.e., we always assign a column exactly one column). Specifically, Item A.1 ensures that a column corresponding to a variable $x_j$ or $a_j$ is never assigned more than one color in Step 2. Similarly, Item A.2 and Item A.3 ensures that each node in the revised implication graph is assigned exactly one color, and that every pair of columns connected with associated nodes in the revised implication is assigned different colors.

We now show that this coloring is equitable (i.e., the sum of the columns colored red minus the sum of the columns colored blue differ by at most one for each row). We denote the sum of the columns colored red minus the sum of the columns colored blue for each row as the *row sum*. Since each row corresponds to a single constraint, we show this result for each constraint class separately:

- $K$**-hot Constraints**: This follows directly from Step 1 and Step 2.

- **Unit-Directional Linkage Constraints**: Consider a generic sub-matrix of $Z$. If the row corresponding to a directional linkage constraint $i$ that operates on index $j$ and $k$ has columns corresponding to $x_j$ and $a_j$ ($x_k$ and $a_k$) in the the submatrix, Step 1 ensures the net sum for columns for variables corresponding to $j$ ($k$) is 0. If after Step 1, there is exactly one remaining uncolored column in the row of the submatrix then the entire row sum is $\pm 1$. If there are two uncolored columns then it must correspond to one variable (i.e., $x_j$ or $a_j$) from index $j$ and one from index $k$. Step 3 ensures that these are assigned different colors. Thus in all cases, the row sum is 0 or $\pm 1$.

- **Integer Bound Constraints**: Every corresponding to an integer bound constraint in the submatrix includes columns correspond to either $x_j$ AND $a_j$, $x_j$, or $a_j$. In the first case, Step 1 ensures that the row sum is 0. In the latter case, the row sum will be $\pm 1$ depending on the color selected in Step 4.

□

We now prove the full result of Theorem 3.2.

*Proof.* By Lemma C.1, under Assumptions A.1- A.3 the linear recourse polyhedron is totally unimodular. This means that every extreme point of the polyhedron is integral and corresponds to feasible *integer* vectors $\mathbf{x}, \mathbf{a}$.

The REP with linear recourse constraints corresponds to the linear recourse polyhedron with an addition linear constraint (i.e., the linear recourse polyhedron intersected with a half-space). We now show that the REP is feasible iff the linear relaxation of the REP is feasible.

$\Rightarrow$ **(REP is feasible implies the linear relaxation of the REP is feasible)**   This follows from the fact that the latter problem is a relaxation of the first problem.

$\Leftarrow$ **(The linear relaxation of the REP is feasible implies the REP is feasible)**   Intuitively, this follows from the fact that the REP is the linear recourse polyhedron intersected with a halfspace. If there are any feasible data points in the REP there must be at least one feasible extreme point (which represents a solution to the REP).

Formally, consider a feasible solution to the relaxed REP $\mathbf{v}$ (note that this includes both $\mathbf{x}, \mathbf{a}$ for ease of notation). This solution must be a feasible point in the linear recourse polyhedron (otherwise it would contradict it being a feasible solution). We can represent any feasible point in the linear recourse polyhedron as a convex combination of extreme points of the polyhedron $\{\mu_k\}_k$ by the Minkowski representation theorem:

$$\mathbf{v} = \sum_k \lambda_k \mu_k \quad \text{s.t.} \ \sum_k \lambda_k = 1, \lambda_k \geq 0 \ \forall k$$

Constraint (3a) in the REP implies:

$$b \leq \mathbf{w}^\top \mathbf{v} = \mathbf{w}^\top \left(\sum_k \lambda_k \mu_k\right) = \sum_k \lambda_k \mathbf{w}^\top \mu_k$$

Since $\lambda_k \geq 0$ this means that there is at least one extreme point $\mu_k$ such that $\mathbf{w}^\top \mu_k \geq b$, and thus at least one feasible solution to the discrete REP. □

## D. Practical Guidelines for Selecting Continuous Restrictions for the FCP

Unfortunately, Theorem 3.2 does not hold directly under a broader set of constraints. Consider the following simple example over two features $x_1, x_2$. Let $x_1$ be a binary variable, and $x_2$ be an integer variable bounded between 0 and 10. Add one directional linkage constraint such that increasing $x_1$ by one unit causes $x_2$ to decrease by 10 units. The classifier assigns the desirable outcome if $x_1 + a_1 \geq 0.5$. Consider the region $\mathcal{R} = x_1 \times x_2 = \{0\} \times [5, 10]$. Every individual in the region has *continuous recourse* by setting $a_1 = 0.5$. However, in the discrete version of the REP, which requires $a_1 \in \{0, 1\}$, no individual has recourse because setting $a_1 = 1$ would require violating the bound constraint $x_2 + a_2 \geq 0$.

One strategy to handle general constraints is to restrict a subset of discrete variables in the REP, such that each resulting continuous restriction meets the conditions of Theorem 3.2. In practice, this strategy leads to a small number of continuous restrictions. For example, consider the `heloc` dataset used in our experiments, which has 42 binary features, one integer feature, and 20 non-separable constraints. It contains two constraints that are not linear recourse constraints:

- Actions on `YearsSinceLastDelqTrade`$\leq$`3` will a 3 unit change in `YearsOfAccountHistory`

- Actions on `YearsSinceLastDelqTrade`$\leq$`5` will a 5 unit change in `YearsOfAccountHistory`

Note that these are not linear recourse constraints because a unit change in one variable results in a non-unit change in another. There are over $2^{42}$ possible continuous restrictions which makes solving the full FCP over every restriction computationally intractable. However, restricting `YearsSinceLastDelqTrade`$\leq$`3` and `YearsSinceLastDelqTrade`$\leq$`5` (i.e., fixing each variable to either 0 or 1) transforms the two violating constraints into integer bound constraints. This only generates four continuous restrictions ($2 \times 2$) but still allows the FCP to leverage Theorem 3.2.

This strategy is sufficient for many actionability constraints available with public implementations. For example, all constraints implemented in the `Reach` package (Kothari et al., 2023) (e.g., if-then constraints, directional linkage constraints with non-unit scaling factors) only require restricting one discrete variable. An alternative approach in settings with a large number of non-separable constraints over discrete variables is to enumerate feasible feature vectors for the inter-connected discrete variables and re-formulate them as one feature which represents a categorical encoding over all possible values (similar to the approach detailed in Ustun et al., 2019).

In settings where this is not possible or does not dramatically reduce the number of discrete variables our approach can also be run with a single continuous restriction corresponding to a linear relaxation of the original REP. Solving this *relaxed* FCP can be used to provide weaker guarantees:

- If the relaxed version of the FCP returns a confined box, this box is confined for the discrete version of the problem (any individual without continuous recourse also does not have recourse in the discrete problem).

- If the relaxed FCP certifies that the entire region is confined (i.e., all data points are assigned fixed predictions), then the entire region is guaranteed to be confined in the discrete version of the problem.

- Unfortunately, if the relaxed version of the FCP is infeasible there is no guarantee that the entire region is responsive.

# E. Supplementary Material for Experiments

In this section we present all the actionability constraints for the datasets used in our experiments. Note that the feature space $\mathcal{X}$ for each dataset has the same upper and lower bounds as well as non-separable constraints as the action set. All regions we audit in this paper are represented by boxes with fixed values for immutable features.

## E.1. Actionability Constraints for the `heloc` Dataset

We show a list of all features and their separable actionability constraints in Table 4.

| Name | Type | LB | UB | Actionability | Sign |
|------|------|----|----|---------------|------|
| ExternalRiskEstimate≥40 | {0,1} | 0 | 1 | No | |
| ExternalRiskEstimate≥50 | {0,1} | 0 | 1 | No | |
| ExternalRiskEstimate≥60 | {0,1} | 0 | 1 | No | |
| ExternalRiskEstimate≥70 | {0,1} | 0 | 1 | No | |
| ExternalRiskEstimate≥80 | {0,1} | 0 | 1 | No | |
| YearsOfAccountHistory | $\mathbb{Z}$ | 0 | 50 | No | |
| AvgYearsInFile≥3 | {0,1} | 0 | 1 | Yes | + |
| AvgYearsInFile≥5 | {0,1} | 0 | 1 | Yes | + |
| AvgYearsInFile≥7 | {0,1} | 0 | 1 | Yes | + |
| MostRecentTradeWithinLastYear | {0,1} | 0 | 1 | Yes | |
| MostRecentTradeWithinLast2Years | {0,1} | 0 | 1 | Yes | |
| AnyDerogatoryComment | {0,1} | 0 | 1 | No | |
| AnyTrade120DaysDelq | {0,1} | 0 | 1 | No | |
| AnyTrade90DaysDelq | {0,1} | 0 | 1 | No | |
| AnyTrade60DaysDelq | {0,1} | 0 | 1 | No | |
| AnyTrade30DaysDelq | {0,1} | 0 | 1 | No | |
| NoDelqEver | {0,1} | 0 | 1 | No | |
| YearsSinceLastDelqTrade≤1 | {0,1} | 0 | 1 | Yes | |
| YearsSinceLastDelqTrade≤3 | {0,1} | 0 | 1 | Yes | |
| YearsSinceLastDelqTrade≤5 | {0,1} | 0 | 1 | Yes | |
| NumInstallTrades≥2 | {0,1} | 0 | 1 | Yes | + |
| NumInstallTrades≥3 | {0,1} | 0 | 1 | Yes | + |
| NumInstallTrades≥5 | {0,1} | 0 | 1 | Yes | + |
| NumInstallTrades≥7 | {0,1} | 0 | 1 | Yes | + |
| NumInstallTradesWBalance≥2 | {0,1} | 0 | 1 | Yes | + |
| NumInstallTradesWBalance≥3 | {0,1} | 0 | 1 | Yes | + |
| NumInstallTradesWBalance≥5 | {0,1} | 0 | 1 | Yes | + |
| NumInstallTradesWBalance≥7 | {0,1} | 0 | 1 | Yes | + |
| NumRevolvingTrades≥2 | {0,1} | 0 | 1 | Yes | + |
| NumRevolvingTrades≥3 | {0,1} | 0 | 1 | Yes | + |
| NumRevolvingTrades≥5 | {0,1} | 0 | 1 | Yes | + |
| NumRevolvingTrades≥7 | {0,1} | 0 | 1 | Yes | + |
| NumRevolvingTradesWBalance≥2 | {0,1} | 0 | 1 | Yes | + |
| NumRevolvingTradesWBalance≥3 | {0,1} | 0 | 1 | Yes | + |
| NumRevolvingTradesWBalance≥5 | {0,1} | 0 | 1 | Yes | + |
| NumRevolvingTradesWBalance≥7 | {0,1} | 0 | 1 | Yes | + |
| NetFractionInstallBurden≥10 | {0,1} | 0 | 1 | Yes | + |
| NetFractionInstallBurden≥20 | {0,1} | 0 | 1 | Yes | + |
| NetFractionInstallBurden≥50 | {0,1} | 0 | 1 | Yes | + |
| NetFractionRevolvingBurden≥10 | {0,1} | 0 | 1 | Yes | |
| NetFractionRevolvingBurden≥20 | {0,1} | 0 | 1 | Yes | |
| NetFractionRevolvingBurden≥50 | {0,1} | 0 | 1 | Yes | |
| NumBank2NatlTradesWHighUtilization≥2 | {0,1} | 0 | 1 | Yes | + |

*Table 4.* Separable actionability constraints for the `heloc` dataset.

The non-separable actionability constraints for this dataset include:

1. DirectionalLinkage: Actions on `NumRevolvingTradesWBalance≥2` will induce to actions on `NumRevolvingTrades≥2`. Each unit change in `NumRevolvingTradesWBalance≥2` leads to: 1.00-unit change in `NumRevolvingTrades≥2`

2. DirectionalLinkage: Actions on `NumInstallTradesWBalance≥2` will induce to actions on `NumInstallTrades≥2`. Each unit change in `NumInstallTradesWBalance≥2` leads to: 1.00-unit change in `NumInstallTrades≥2`

3. DirectionalLinkage: Actions on `NumRevolvingTradesWBalance`$\geq$3 will induce to actions on `NumRevolvingTrades`$\geq$3.Each unit change in `NumRevolvingTradesWBalance`$\geq$3 leads to:1.00-unit change in `NumRevolvingTrades`$\geq$3

4. DirectionalLinkage: Actions on `NumInstallTradesWBalance`$\geq$3 will induce to actions on `NumInstallTrades`$\geq$3. Each unit change in `NumInstallTradesWBalance`$\geq$3 leads to:1.00-unit change in `NumInstallTrades`$\geq$3

5. DirectionalLinkage: Actions on `NumRevolvingTradesWBalance`$\geq$5 will induce to actions on `NumRevolvingTrades`$\geq$5.Each unit change in `NumRevolvingTradesWBalance`$\geq$5 leads to:1.00-unit change in `NumRevolvingTrades`$\geq$5

6. DirectionalLinkage: Actions on `NumInstallTradesWBalance`$\geq$5 will induce to actions on `NumInstallTrades`$\geq$5.Each unit change in `NumInstallTradesWBalance`$\geq$5 leads to:1.00-unit change in `NumInstallTrades`$\geq$5

7. DirectionalLinkage: Actions on `NumRevolvingTradesWBalance`$\geq$7 will induce to actions on `NumRevolvingTrades`$\geq$7.Each unit change in `NumRevolvingTradesWBalance`$\geq$7 leads to:1.00-unit change in `NumRevolvingTrades`$\geq$7

8. DirectionalLinkage: Actions on `NumInstallTradesWBalance`$\geq$7 will induce to actions on `NumInstallTrades`$\geq$7.Each unit change in `NumInstallTradesWBalance`$\geq$7 leads to:1.00-unit change in `NumInstallTrades`$\geq$7

9. DirectionalLinkage: Actions on `YearsSinceLastDelqTrade`$\leq$1 will induce to actions on `YearsOfAccountHistory`. Each unit change in `YearsSinceLastDelqTrade`$\leq$1 leads to:-1.00-unit change in `YearsOfAccountHistory`

10. DirectionalLinkage: Actions on `YearsSinceLastDelqTrade`$\leq$3 will induce to actions on `YearsOfAccountHistory`. Each unit change in `YearsSinceLastDelqTrade`$\leq$3 leads to:-3.00-unit change in `YearsOfAccountHistory`

11. DirectionalLinkage: Actions on `YearsSinceLastDelqTrade`$\leq$5 will induce to actions on `YearsOfAccountHistory`. Each unit change in `YearsSinceLastDelqTrade`$\leq$5 leads to:-5.00-unit change in `YearsOfAccountHistory`

12. ThermometerEncoding: Actions on [`YearsSinceLastDelqTrade`$\leq$1, `YearsSinceLastDelqTrade`$\leq$3, `YearsSinceLastDelqTrade`$\leq$5] must preserve thermometer encoding., which can only decrease.Actions can only turn off higher-level dummies that are on, where `YearsSinceLastDelqTrade`$\leq$1 is the lowest-level dummy and `YearsSinceLastDelqTrade`$\leq$5 is the highest-level-dummy.

13. ThermometerEncoding: Actions on [`MostRecentTradeWithinLast2Years`, `MostRecentTradeWithinLastYear`] must preserve thermometer encoding.

14. ThermometerEncoding: Actions on [`AvgYearsInFile`$\geq$3, `AvgYearsInFile`$\geq$5, `AvgYearsInFile`$\geq$7] must preserve thermometer encoding., which can only increase.Actions can only turn on higher-level dummies that are off, where `AvgYearsInFile`$\geq$3 is the lowest-level dummy and `AvgYearsInFile`$\geq$7 is the highest-level-dummy.

15. ThermometerEncoding: Actions on [`NetFractionRevolvingBurden`$\geq$10, `NetFractionRevolvingBurden`$\geq$20, `NetFractionRevolvingBurden`$\geq$50] must preserve thermometer encoding., which can only decrease.Actions can only turn off higher-level dummies that are on, where `NetFractionRevolvingBurden`$\geq$10 is the lowest-level dummy and `NetFractionRevolvingBurden`$\geq$50 is the highest-level-dummy.

16. ThermometerEncoding: Actions on [`NetFractionInstallBurden`$\geq$10, `NetFractionInstallBurden`$\geq$20, `NetFractionInstallBurden`$\geq$50] must preserve thermometer encoding., which can only decrease.Actions can only turn off higher-level dummies that are on, where `NetFractionInstallBurden`$\geq$10 is the lowest-level dummy and `NetFractionInstallBurden`$\geq$50 is the highest-level-dummy.

17. ThermometerEncoding: Actions on [NumRevolvingTradesWBalance$\geq$2, NumRevolvingTradesWBalance$\geq$3, NumRevolvingTradesWBalance$\geq$5, NumRevolvingTradesWBalance$\geq$7] must preserve thermometer encoding., which can only decrease.Actions can only turn off higher-level dummies that are on, where NumRevolvingTradesWBalance$\geq$2 is the lowest-level dummy and NumRevolvingTradesWBalance$\geq$7 is the highest-level-dummy.

18. ThermometerEncoding: Actions on [NumRevolvingTrades$\geq$2, NumRevolvingTrades$\geq$3, NumRevolvingTrades$\geq$5, NumRevolvingTrades$\geq$7] must preserve thermometer encoding., which can only decrease.Actions can only turn off higher-level dummies that are on, where NumRevolvingTrades$\geq$2 is the lowest-level dummy and NumRevolvingTrades$\geq$7 is the highest-level-dummy.

19. ThermometerEncoding: Actions on [NumInstallTradesWBalance$\geq$2, NumInstallTradesWBalance$\geq$3, NumInstallTradesWBalance$\geq$5, NumInstallTradesWBalance$\geq$7] must preserve thermometer encoding., which can only decrease.Actions can only turn off higher-level dummies that are on, where NumInstallTradesWBalance$\geq$2 is the lowest-level dummy and NumInstallTradesWBalance$\geq$7 is the highest-level-dummy.

20. ThermometerEncoding: Actions on [NumInstallTrades$\geq$2, NumInstallTrades$\geq$3, NumInstallTrades$\geq$5, NumInstallTrades$\geq$7] must preserve thermometer encoding., which can only decrease.Actions can only turn off higher-level dummies that are on, where NumInstallTrades$\geq$2 is the lowest-level dummy and NumInstallTrades$\geq$7 is the highest-level-dummy.

## E.2. Actionability Constraints for the `givemecredit` Dataset

We present a list of all features and their separable actionability constraints in Table 5.

| Name | Type | LB | UB | Actionability | Sign |
|------|------|-----|-----|---------------|------|
| Age | $\mathbb{Z}$ | 21 | 103 | No | |
| NumberOfDependents | $\mathbb{Z}$ | 0 | 20 | No | |
| DebtRatio$\geq$1 | $\{0,1\}$ | 0 | 1 | Yes | + |
| MonthlyIncome$\geq$3K | $\{0,1\}$ | 0 | 1 | Yes | + |
| MonthlyIncome$\geq$5K | $\{0,1\}$ | 0 | 1 | Yes | + |
| MonthlyIncome$\geq$10K | $\{0,1\}$ | 0 | 1 | Yes | + |
| CreditLineUtilization$\geq$10 | $\{0,1\}$ | 0 | 1 | Yes | |
| CreditLineUtilization$\geq$20 | $\{0,1\}$ | 0 | 1 | Yes | |
| CreditLineUtilization$\geq$50 | $\{0,1\}$ | 0 | 1 | Yes | |
| CreditLineUtilization$\geq$70 | $\{0,1\}$ | 0 | 1 | Yes | |
| CreditLineUtilization$\geq$100 | $\{0,1\}$ | 0 | 1 | Yes | |
| AnyRealEstateLoans | $\{0,1\}$ | 0 | 1 | Yes | + |
| MultipleRealEstateLoans | $\{0,1\}$ | 0 | 1 | Yes | + |
| AnyCreditLinesAndLoans | $\{0,1\}$ | 0 | 1 | Yes | + |
| MultipleCreditLinesAndLoans | $\{0,1\}$ | 0 | 1 | Yes | + |
| HistoryOfLatePayment | $\{0,1\}$ | 0 | 1 | No | |
| HistoryOfDelinquency | $\{0,1\}$ | 0 | 1 | No | |

*Table 5.* Separable actionability constraints for the `givemecredit` dataset.

The non-separable actionability constraints for this dataset include:

1. ThermometerEncoding: Actions on [`MonthlyIncome`$\geq$`3K`, `MonthlyIncome`$\geq$`5K`, `MonthlyIncome`$\geq$`10K`] must preserve thermometer encoding., which can only increase.Actions can only turn on higher-level dummies that are off, where `MonthlyIncome`$\geq$`3K` is the lowest-level dummy and `MonthlyIncome`$\geq$`10K` is the highest-level-dummy.

2. ThermometerEncoding: Actions on [`CreditLineUtilization`$\geq$`10`, `CreditLineUtilization`$\geq$`20`, `CreditLineUtilization`$\geq$`50`, `CreditLineUtilization`$\geq$`70`, `CreditLineUtilization`$\geq$`100`] must preserve thermometer encoding., which can only decrease.Actions can only turn off higher-level dummies that are on, where `CreditLineUtilization`$\geq$`10` is the lowest-level dummy and `CreditLineUtilization`$\geq$`100` is the highest-level-dummy.

3. ThermometerEncoding: Actions on [`AnyRealEstateLoans`, `MultipleRealEstateLoans`] must preserve thermometer encoding., which can only decrease.Actions can only turn off higher-level dummies that are on, where `AnyRealEstateLoans` is the lowest-level dummy and `MultipleRealEstateLoans` is the highest-level-dummy.

4. ThermometerEncoding: Actions on [`AnyCreditLinesAndLoans`, `MultipleCreditLinesAndLoans`] must preserve thermometer encoding., which can only decrease.Actions can only turn off higher-level dummies that are on, where `AnyCreditLinesAndLoans` is the lowest-level dummy and `MultipleCreditLinesAndLoans` is the highest-level-dummy.

### E.3. Actionability Constraints for the `twitterbot` Dataset

We present a list of all features and their separable actionability constraints in Table 6.

| Name | Type | LB | UB | Actionability | Sign |
|---|---|---|---|---|---|
| SourceAutomation | $\{0,1\}$ | 0 | 1 | No | |
| SourceOther | $\{0,1\}$ | 0 | 1 | No | |
| SourceBranding | $\{0,1\}$ | 0 | 1 | No | |
| SourceMobile | $\{0,1\}$ | 0 | 1 | No | |
| SourceWeb | $\{0,1\}$ | 0 | 1 | No | |
| SourceApp | $\{0,1\}$ | 0 | 1 | No | |
| FollowerFriendRatio$\geq$1 | $\{0,1\}$ | 0 | 1 | No | |
| FollowerFriendRatio$\geq$10 | $\{0,1\}$ | 0 | 1 | No | |
| FollowerFriendRatio$\geq$100 | $\{0,1\}$ | 0 | 1 | No | |
| FollowerFriendRatio$\geq$1000 | $\{0,1\}$ | 0 | 1 | No | |
| FollowerFriendRatio$\geq$10000 | $\{0,1\}$ | 0 | 1 | No | |
| FollowerFriendRatio$\geq$100000 | $\{0,1\}$ | 0 | 1 | No | |
| AgeOfAccountInDays$\geq$365 | $\{0,1\}$ | 0 | 1 | Yes | |
| AgeOfAccountInDays$\geq$730 | $\{0,1\}$ | 0 | 1 | Yes | |
| UserReplied$\geq$10 | $\{0,1\}$ | 0 | 1 | Yes | |
| UserReplied$\geq$100 | $\{0,1\}$ | 0 | 1 | Yes | |
| UserFavourited$\geq$1000 | $\{0,1\}$ | 0 | 1 | Yes | |
| UserFavourited$\geq$10000 | $\{0,1\}$ | 0 | 1 | Yes | |
| UserRetweeted$\geq$1 | $\{0,1\}$ | 0 | 1 | Yes | |
| UserRetweeted$\geq$10 | $\{0,1\}$ | 0 | 1 | Yes | |
| UserRetweeted$\geq$100 | $\{0,1\}$ | 0 | 1 | Yes | |

*Table 6.* Separable actionability constraints for the `Twitterbot` dataset.

The non-separable actionability constraints for this dataset include:

1. ThermometerEncoding: Actions on [FollowerFriendRatio$\geq$1, FollowerFriendRatio$\geq$10, FollowerFriendRatio$\geq$100, FollowerFriendRatio$\geq$1000, FollowerFriendRatio$\geq$10000, FollowerFriendRatio$\geq$100000] must preserve thermometer encoding.

2. ThermometerEncoding: Actions on [AgeOfAccountInDays$\geq$365, AgeOfAccountInDays$\geq$730] must preserve thermometer encoding.

3. ThermometerEncoding: Actions on [UserReplied$\geq$10, UserReplied$\geq$100] must preserve thermometer encoding.

4. ThermometerEncoding: Actions on [UserFavourited$\geq$1000, UserFavourited$\geq$10000] must preserve thermometer encoding.

5. ThermometerEncoding: Actions on [UserRetweeted$\geq$1, UserRetweeted$\geq$10, UserRetweeted$\geq$100] must preserve thermometer encoding.

### E.4. Computing Infrastructure

We run all experiments on a personal computer with an Apple M1 Pro chip and 32 GB of RAM. All MILP and MIQCP problems were solved using Gurobi 9.0 (Achterberg, 2019) with default settings.

### E.5. Additional Results

Table 7 shows an extended version of our main results that include three additional metrics:

- **Realized Blindspot Rate**: The fraction of total regions that are predicted to be responsive but contain individuals with fixed predictions in the test dataset.

| Dataset | Metrics | PointWise | | | ReVer |
| --- | --- | --- | --- | --- | --- |
| | | Data | Region | Score | |
| heloc
$n = 5842$  $d = 43$
$|\Omega| = 155$  $p = 22.2\%$
FICO (2018) | Certifies Responsive | — | — | — | 54.2% |
| | Outputs Responsive | 91.6% | 66.5% | 71.0% | 54.2% |
| | ↳ Blindspot | 37.4% | 12.3% | 16.8% | **0.0%** |
| | ↳ Realized Blindspot | 1.9% | **0.0%** | **0.0%** | **0.0%** |
| | Certifies Confined | — | — | — | 0.0% |
| | Outputs Confined | 0.6% | 0.0% | 0.0% | 0.0% |
| | ↳ Loophole | 0.6% | **0.0%** | **0.0%** | **0.0%** |
| | ↳ Realized Loophole | **0.0%** | **0.0%** | **0.0%** | **0.0%** |
| | Comp. Time (s) | 0.05(0.0) | 0.74(0.1) | 0.02(0.0) | 4.67(3.6) |
| givemecredit
$n = 120,268$  $d = 23$
$|\Omega| = 715$  $p = 7.4\%$
Kaggle (2011) | Certifies Responsive | — | — | — | 60.1% |
| | Outputs Responsive | 72.2% | 60.1% | 62.9% | 60.1% |
| | ↳ Blindspot | 12.0% | **0.0%** | 2.8% | **0.0%** |
| | ↳ Realized Blindspot | 3.1% | **0.0%** | 1.0% | **0.0%** |
| | Certifies Confined | — | — | — | 18.3% |
| | Outputs Confined | 19.4% | 19.2% | 19.2% | 18.3% |
| | ↳ Loophole | 1.1% | 0.8% | 0.8% | **0.0%** |
| | ↳ Realized Loophole | 0.3% | **0.0%** | **0.0%** | **0.0%** |
| | Comp. Time (s) | 0.02(0.1) | 0.32(0.1) | 0.01(0.0) | 0.13(0.1) |
| twitterbot
$n = 1438$  $d = 21$
$|\Omega| = 20$  $p = 55.3\%$
Gilani et al. (2016) | Certifies Responsive | — | — | — | 25.0% |
| | Outputs Responsive | 40.0% | 25.0% | 25.0% | 25.0% |
| | ↳ Blindspot | 15.0% | **0.0%** | **0.0%** | **0.0%** |
| | ↳ Realized Blindspot | 15.0% | **0.0%** | **0.0%** | **0.0%** |
| | Certifies Confined | — | — | — | 5.0% |
| | Outputs Confined | 25.0% | 25.0% | 25.0% | 5.0% |
| | ↳ Loophole | 20.0% | 20.0% | 20.0% | **0.0%** |
| | ↳ Realized Loophole | **0.0%** | **0.0%** | **0.0%** | **0.0%** |
| | Comp. Time (s) | 0.02(0.1) | 0.36(0.1) | 0.01(0.0) | 0.07(0.2) |

*Table 7.* Overview of results for all datasets, regions, and methods. For each dataset, we include the number of regions we audit ($|\Omega|$), and the fraction of data points with fixed predictions ($p$).

- **Realized Loophole Rate**: The fraction of total regions that are predicted to be confined but contain individuals with recourse in the test dataset.

- **Computation Time**: The average computation time in seconds over all regions for each dataset.

As expected, the *realized* blindspot and loophole rates are lower than the true blindspot and loophole rate. Recourse verification over regions safeguards against rare events (i.e., new individuals with fixed predictions that were not part of the training dataset), which makes the risks less likely to be realized over a small test dataset. However in real-world applications, models are typically deployed and make predictions on datasets much larger than its original training dataset — increasing the probability of blindspots and loopholes being realized.

Our extended results also highlight that our approach runs incredibly quickly across all three datasets, auditing a region in under 5 seconds on average. In two out of three datasets, our approach is quicker than the Region which runs pointwise recourse on 100 data points.

# F. On Out-of-Sample Robustness

Auditing recourse over regions, rather than individuals, allows practitioners to find individuals with fixed predictions beyond a training dataset and is robust to distribution shifts. We demonstrate this capability by considering *realized blindspots*, regions that are predicted to be responsive but contain individuals with fixed predictions in the test distribution. We evaluate the performance of the Data baseline, which certifies recourse by looking at individuals within a training dataset, in two regimes: (1) where the test distribution is the same as the train distribution of data, (2) where the train distribution is more likely to include individuals predicted to receive the desirable outcome. We simulate this distribution shift by training a logistic regression classifier on the entire dataset that predicts the likelihood of receiving the desirable outcome. We then construct the training dataset by sampling individuals with a probability proportional to their predicted score in the linear classifier using a soft-max function with a temperature of 1. In Figure 3 we plot the realized blindspot rate for the Data baseline in all datasets with and without distribution shift. Our results show that *even under the same distribution* the Data baseline can fail to catch instances of fixed predictions in the test dataset. This problem is further exacerbated by distribution shift, with the realized blindspot rate increasing across all three datasets. Note that by design ReVer has 0 realized blindspots because the regions themselves remain fixed under both train and test. Overall, these results highlight the importance of auditing regions as a tool to robustly foresee future fixed predictions, even under distribution shift.

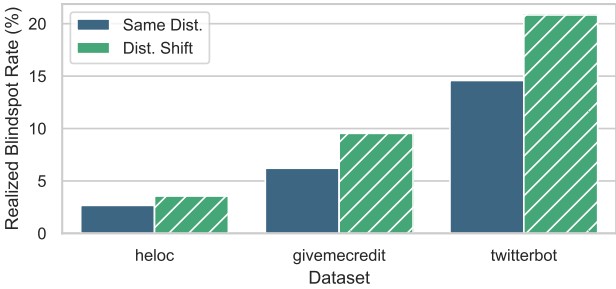

*Figure 3.* Realized blindspot rate for Data baseline with and without a distribution shift. Realized blindspot rate is the percentage of regions predicted to be responsive that contain individuals with fixed predictions in test dataset.

# G. Supplementary Material for Case Study on Pennsylvania SRAI

In this section we present all the actionability constraints for the Pennsylvania SRAI. Note that the feature space $\mathcal{X}$ for this setting has the same upper and lower bounds as well as non-separable constraints as the action set. For this case study, we audit a box region that contains any possible offender.

## G.1. Actionability Constraints for the SRAI

We show a list of all features and their separable actionability constraints in Table 8.

| Name | Type | LB | UB | Actionability | Sign |
|------|------|----|----|---------------|------|
| GenderMale | $\{0,1\}$ | 0 | 1 | No | |
| GenderFemale | $\{0,1\}$ | 0 | 1 | No | |
| AgeUnder21: | $\{0,1\}$ | 0 | 1 | No | |
| Age21-25 | $\{0,1\}$ | 0 | 1 | No | |
| Age26-29 | $\{0,1\}$ | 0 | 1 | No | |
| Age30-39 | $\{0,1\}$ | 0 | 1 | No | |
| Age40-49 | $\{0,1\}$ | 0 | 1 | No | |
| Age50+ | $\{0,1\}$ | 0 | 1 | No | |
| CurrentConvictionMurder | $\{0,1\}$ | 0 | 1 | Yes | |
| CurrentConvictionPerson-Felony | $\{0,1\}$ | 0 | 1 | Yes | |
| CurrentConvictionPerson-Misd. | $\{0,1\}$ | 0 | 1 | Yes | |
| CurrentConvictionSex-Felony | $\{0,1\}$ | 0 | 1 | Yes | |
| CurrentConvictionSex-Misd. | $\{0,1\}$ | 0 | 1 | Yes | |
| CurrentConvictionBurglary | $\{0,1\}$ | 0 | 1 | Yes | |
| CurrentConvictionProperty-Felony | $\{0,1\}$ | 0 | 1 | Yes | |
| CurrentConvictionProperty-Misd. | $\{0,1\}$ | 0 | 1 | Yes | |
| CurrentConvictionDrug-Felony | $\{0,1\}$ | 0 | 1 | Yes | |
| CurrentConvictionDrug-Misd | $\{0,1\}$ | 0 | 1 | Yes | |
| CurrentConvictionPublic-Admin. | $\{0,1\}$ | 0 | 1 | Yes | |
| CurrentConvictionPublic-Order. | $\{0,1\}$ | 0 | 1 | Yes | |
| CurrentConvictionFirearms | $\{0,1\}$ | 0 | 1 | Yes | |
| CurrentConvictionOther Weapons | $\{0,1\}$ | 0 | 1 | Yes | |
| CurrentConvictionOther | $\{0,1\}$ | 0 | 1 | Yes | |
| NumPriorConvictionsNone | $\{0,1\}$ | 0 | 1 | Yes | |
| NumPriorConvictions1 | $\{0,1\}$ | 0 | 1 | Yes | |
| NumPriorConvictions2-3 | $\{0,1\}$ | 0 | 1 | Yes | |
| NumPriorConvictions4-5 | $\{0,1\}$ | 0 | 1 | Yes | |
| NumPriorConvictions5+ | $\{0,1\}$ | 0 | 1 | Yes | |
| PriorConvictionPerson/Sex | $\{0,1\}$ | 0 | 1 | Yes | |
| PriorConvictionProperty | $\{0,1\}$ | 0 | 1 | Yes | |
| PriorConvictionDrug | $\{0,1\}$ | 0 | 1 | Yes | |
| PriorConvictionPublicOrder | $\{0,1\}$ | 0 | 1 | Yes | |
| PriorConvictionPublicAdmin | $\{0,1\}$ | 0 | 1 | Yes | |
| PriorConvictionDUI | $\{0,1\}$ | 0 | 1 | Yes | |
| PriorConvictionFirearm | $\{0,1\}$ | 0 | 1 | Yes | |
| MultipleCurrentConvictions | $\{0,1\}$ | 0 | 1 | Yes | |
| PriorJuvenileAdjudication | $\{0,1\}$ | 0 | 1 | Yes | |

*Table 8.* Separable actionability constraints for the Pennsylvania SRAI.

The non-separable actionability constraints for this dataset include:

1. OneHotEncoding: Actions on [AgeUnder21, Age21-25, Age26-29, Age30-39, Age40-49, Age50+] must preserve one-hot encoding of Age. Exactly 1 of [AgeUnder21, Age21-25, Age26-29, Age30-39, Age40-49, Age50+] must be TRUE

2. OneHotEncoding: Actions on [GenderMale, GenderFemale] must preserve one-hot encoding of Gender. Exactly 1 of [GenderMale, GenderFemale] must be TRUE

3. Actions on [NumPriorConvictionsNone, NumPriorConvictions2-3, NumPriorConvictions4-5, NumPriorConvictions5+] must preserve one-hot encoding of NumPriorConvictions. Exactly 1

of [NumPriorConvictionsNone, NumPriorConvictions2-3, NumPriorConvictions4-5, NumPriorConvictions5+] must be TRUE

4. LogicalConstraint: If NumPriorConvictionsNone is True then any PriorConviction] variable must be False.

