# OpenReview forum: "Understanding Fixed Predictions via Confined Regions"
_ICML.cc/2025/Conference — ICML 2025 poster_

### Official Review · Reviewer_eC8e · 2025-02-25

**Overall Recommendation:** 3

**Summary:**

This paper introduces the ReVer method, designed to detect fixed predictions in machine learning models by identifying confined regions in the feature space. The approach involves formulating the Region Recourse Verification Problem (RVP), approximating confined regions using bounding boxes, generating confined bounding boxes, and verifying their confinement through the Farkas Certificate Problem (FCP). Experimental results demonstrate the method's effectiveness in accurately identifying confined regions.

## update after rebuttal
The authors clarified that the key assumption underlying their method is generally valid in real-world scenarios(datasets), addressing my main concern. Thus, I maintain a positive rating.

**Claims And Evidence:**

- The claim that “The existing point-wise verification methods fail to discover confined regions” is supported by the experimental results of baselines in Section 4.
- The claim that “The proposed ReVer method is able to verify and find confined regions ” is supported by the introduced theoretical guarantees based on Farkas’ lemma and experimental evidence.
- The claim that “The proposed ReVer method enables data-free auditing, suitable for privacy-sensitive scenarios” is supported by the case study on the SRAI.

**Essential References Not Discussed:**

The paper thoroughly discusses previous work and delineates distinctions. While no critical references seem to be missing, the possibility of omissions cannot be entirely ruled out as I am not very familiar with this topic.

**Experimental Designs Or Analyses:**

The experimental design is sound, encompassing diverse applications and selecting appropriate baselines, and the results support the paper’s claims. Additionally, the SRAI case study highlights the method's practicality in scenarios with unavailable data. However, the experiments may lack analysis of varying model complexities, such as the dimensionality of feature spaces (coefficients $w$).

**Methods And Evaluation Criteria:**

The proposed method and evaluation criteria are well-aligned with the defined problem and application scenarios. The approach effectively addresses the fixed prediction issue, while the datasets and evaluation metrics adequately validate its performance.

**Other Comments Or Suggestions:**

Missing punctuation in sentences and formulas, e.g., Line 145. Please review thoroughly.

**Other Strengths And Weaknesses:**

**Strengths**:
- S1: The topic is impactful, addressing potential blind spots and loopholes in real-world scenarios.
- S2: The introduction of Farkas Certificate Problem (FCP) and Mixed Integer Quadratically Constrained Programming (MIQCP) tools provides strong theoretical guarantees.
- S3- Practicality of the method is evident, especially in "detecting harms before deployment" and "data-free auditing."

**Weaknesses**:
- W1: The inclusion of certain baseline methods mentioned in the Introduction, such as [22], could enhance the comprehensiveness of the study.
- W2: The performance under varying parameter dimensions remains unclear. It is uncertain whether the proposed method consistently outperforms baselines as (linear) model scales.

**Questions For Authors:**

**Questions**:
- Q1: How does restricting the response variable $y$ to binary values influence the problem formulation? Can the method generalize to cases where $y$ has $N \geq 2$ values? For instance, if transitions between undesired outcomes exist under certain actions but no action leads to the desired outcome, is this still a fixed prediction?
- Q2: In Figure 1, is the range of $x_1$ $[0, 5)$ or $[0, 5]$?
- Q3: On Line 220, the authors claim, "a box is confined if and only if every continuous restriction of the REP is infeasible." Could you clarify the insight behind this conclusion?
- Q4: For discrete variables, the method relies on assumptions of linear recourse constraints. Were these assumptions met in the experiments (Section 4) and case study (Section 5)?

**Relation To Broader Scientific Literature:**

Building on existing work in algorithmic recourse, this paper pioneers a region-level rather than individual-level perspective on recourse verification, addressing a gap in region-level recourse analysis. Besides, unlike prior studies focusing more on action spaces, this work targets confined regions in the feature space.

**Theoretical Claims:**

I have not verified the correctness of the theoretical claims presented in the paper as I am not familiar with the used techniques.

---

> ### Author Rebuttal · Authors · 2025-04-01
>
> Thank you for your time and feedback! We really appreciate you highlighting that our work address a topic that is “impactful” where the “practicality [of our approach]is evident”, our MIQCP approach provides “strong theoretical guarantees”, and that our “experimental design is sound, encompassing diverse applications and selecting appropriate baselines” which “supports the papers’ claims”.
>
> We also appreciate your comments and concerns which have helped us refine the paper, we address them below:
> > The inclusion of certain baseline methods mentioned in the Introduction, such as [22], could enhance the comprehensiveness of the study.
>
> We solve a MILP to check whether a given data point has recourse (specifically we solve the REP for the given point x where B(u,l) = x). This can be viewed as an extension of AR [1] to handle more diverse actionability constraints, or a special case of Reach [2] tailored to linear models. Note that this MILP provably verifies recourse (or its absence) so would provide the same response as [2], but runs much faster (<1s for the MILP vs. minutes for [22]). Note that any other algorithmic recourse approach (e.g., DiCE [3]) would only provide weaker guarantees than this baseline as they cannot formally verify recourse. We use this approach for both our pointwise verification, as well as to compute p (i.e., by running the approach on the entire dataset). We’ve clarified this in the revised draft!
>
> > The performance under varying parameter dimensions remains unclear. It is uncertain whether the proposed method consistently outperforms baselines as (linear) model scales.
>
> Great point! In our paper we benchmark our approach on standard datasets in the algorithmic recourse community (e.g., from [1], [2]) which capture realistic parameter distributions. We expect our approach to actually improve more over baseline approaches as the dimensionality of the data increases. For some intuition, as the dimension increases, the odds of having data (or generating samples) that cover the entire region decreases (more space to cover!). In practice, this means blindspots are more likely to occur. We see this in our experimental results where the baselines perform the worst on heloc, which has the largest dimensionality.
>
> > How does restricting the response variable y to binary values influence the problem formulation? Can the method generalize to cases where y has $N \geq 2$  values? For instance, if transitions between undesired outcomes exist under certain actions but no action leads to the desired outcome, is this still a fixed prediction?
>
> Thanks for asking! Recourse is generally defined as having a ‘desired outcome’ (e.g., loan is accepted) which naturally defines a binary problem (‘desired outcome’ vs. not) even if y has multiple values. With this definition in mind, transitions between undesired outcomes but no action leading to the desired outcome would count as a fixed prediction.
>
> > In Figure 1, is the range of x_1 [0,5) or [0,5]?
>
> Thanks for catching this typo! The range of x_1 should be [0,5).
>
> > On Line 220, the authors claim, "a box is confined if and only if every continuous restriction of the REP is infeasible." Could you clarify the insight behind this conclusion?
>
> Recall that a continuous restriction is a ‘smaller’ version of the original MILP where all discrete variables have been fixed to a specific value (e.g., leaving just a continuous problem). If a box is confined, that means that there is no data point that has recourse - this implies that whatever ways we fix the discrete variables in the REP (e.g., setting elements of x and a) the resulting continuous problem has to be infeasible (otherwise we have found a contradiction to the box being confined). Similarly in the reverse direction, if there is a specific continuous restriction that is feasible then we can use the solution to the continuous restriction to find a data point with recourse in the region (proving the box isn’t confined). We’ve added an extra sentence explaining this in the revised draft.
>
> > For discrete variables, the method relies on assumptions of linear recourse constraints. Were these assumptions met in the experiments (Section 4) and case study (Section 5)?
>
> Almost! The case study, twitterbot, and givemecredit datasets meet all the assumptions for linear recourse constraints. The heloc dataset does not, but we can provably certify recourse by only considering 4 continuous restrictions in the RVP. We go into this in greater depth in Appendix D in the supplementary materials.
>
> Thanks again for your time and feedback! Please let us know if you have any additional questions.
>
> [1] Actionable Recourse in Linear Classification. Ustun et al. ACM FAT* (2019)
>
> [2] Prediction without Preclusion: Recourse Verification with Reachable Sets. Kothari at al. ICLR (2025)
>
> [3] Explaining machine learning classifiers through diverse counterfactual explanations. Mothilal et al. FAccT (2020).

---

> > ### Comment · Reviewer_eC8e · 2025-04-04
> >
> > Thank you for your rebuttal, which addressed my concerns. I will maintain my positive score.

---

### Official Review · Reviewer_eTLW · 2025-03-06

**Overall Recommendation:** 3

**Summary:**

The paper proposes a method to check whether regions of input features are *responsive* (all individuals in the region have recourse), *confined* (all individuals have no recourse) or neither.

**Claims And Evidence:**

- The authors claim that they “introduce a new approach to formally verify recourse over entire regions of the feature space”. It is unclear throughout the text if they introduce a new method to verify recourse over regions or if they introduce the problem itself. In any case, it should be better justified in the related work how this is novel.
- In contributions, the paper also claims to evaluate their approach. I would reword this contribution, as evaluation is part of the process for justifying one’s approach. Using the sentence “pointwise verification approaches fail \[…\]” would fit more this part of the text. Anyways, I am concerned about how the evidence of this last claim because there is not enough evidence of what pointwise verification approaches they use (see Methods and Evidence, point 2.).
- The paper claims that their approach is fast but do not give enough evidence of comparison nor how the time would scale with regions or datapoints.

**Essential References Not Discussed:**

Not sure.

**Experimental Designs Or Analyses:**

- I have not reproduced their experiments.
- I am concerned about the experiment section only showing the results for one run of the experiment. Several runs would add robustness to the claims. There is no evidence in the paper that the results have been run several times but in the Appendix E they show “average computation time”, which is needs several runs. This is confusing.
- The experiments also do not show results for changing the way the regions are calculated. This would add soundness to the experiments.
- I am also concerned that they have no baseline to compare with a purely region verification. Adding a synthetic experiment where they can calculate the closed-form results beforehand would add validity to the method (like what they did in Figure 1).

**Methods And Evaluation Criteria:**

- Regarding the strategies for generating data points for the baselines, I am unsure why *Score* is relevant for comparison (*Data* includes *Score*).  A discussion of how they all are relevant would be useful to compare approaches when discussing the results.
- What pointwise method is used as baseline i.e. how do you calculate if a point is a *fixed prediction*? How do does this differ from when you calculate the parameter of the dataset $p$?

**Other Comments Or Suggestions:**

Citations are not in the usual ICML format, but I don't know whether that is a problem.

**Other Strengths And Weaknesses:**

-   Weaknesses

    -   Clarity, coherence and cohesion of the papers. The paper is
        unorganized (e.g. no subsections), information is sometimes
        repeated and key information is omitted. Giving the name when
        concepts are first mentioned in the introduction (*RVP*,
        ReVer...) would add a lot of clarity to the text .

    -   Would this approach work for more complex models? Is it possible
        to see the results in such models?

    -   See **Methods and evidence**

    -   See **Experimental Designs or Analyses**

-   Strengths

    -   The paper proposes an original approach to recourse that could
        be very useful due to their properties.

**Questions For Authors:**

See Weaknesses.

I would be willing to change my score if the paper is given more explicit structure and
improved clarity, coherence and cohesion (readability). Furthermore, all
experimental concerns would have to be addressed.

**Relation To Broader Scientific Literature:**

The paper (apparently – see Claims and Evidence) proposes a new paradigm to verify recourse in the input feature space.

**Theoretical Claims:**

I have not checked the correctness of the mathematical derivations

---

> ### Author Rebuttal · Authors · 2025-04-01
>
> Thank you for your time and feedback! We appreciate your comments about the paper, which we address below:
>
> > What pointwise method is used as baseline…
>
> We solve a MILP to check whether a given data point has recourse (i.e., solve the REP for the given point x where B(u,l) = x). This can be viewed as an extension of AR [1] to handle more diverse actionability constraints. Note that this MILP provably verifies recourse (or its absence) so would provide the same response as [2], but runs much faster (<1s for the MILP vs. minutes for [2]). Any other recourse approach (e.g., DiCE [3]) would provide weaker guarantees than our baseline as they cannot formally verify recourse. We use this for pointwise verification and to compute p. We’ve clarified this in the new draft!
>
> > It is unclear throughout the text if they introduce a new method to verify recourse over regions or if they introduce the problem itself…
>
> Thanks for pointing this out. The answer is both! This paper both introduces the problem of verifying recourse over regions and introduces the first method to solve it. We tried to articulate this in our introduction and the related work. We've revised these accordingly but let us know if it isn't clear and we can fix it.
>
> > The paper claims that their approach is fast but do not give enough evidence of comparison nor how the time would scale with regions or datapoints.
>
> We compare the speed of our approach to pointwise baselines in Tab. 7 which shows that our approach runs in seconds on all three of our baseline datasets (which are in line, with respect to size, with existing work in recourse e.g., [1]- [3]).
> We also think there might be a small misunderstanding - ReVeR is called once per region and so its computation time is independent of the number of data points and only scales with the dimension of the classifier which is an improvement over pointwise approaches which scale with the number of points.
>
> > I am unsure why Score is relevant for comparison…
>
> Score tests the point in the entire region (i.e., looks at the geometry of the region) with the highest/lowest score as opposed to any data sampled from the region. Consider a simple example of a 1D region $x \in [0, 10]$. Score tests the points $x = 0$, $x = 10$ (which may not be present in any existing dataset), whereas Data only looks at available training data. We include Score to test the intuition that the ‘worse-off’ member of the region has to take the largest amount of action to change their prediction and thus is more likely to have a fixed prediction. We include additional discussion in the revised draft.
>
> > I am concerned about the experiment section only showing the results for one run of the experiment…
>
> We think there’s been a misunderstanding! For each dataset we evaluate multiple regions (described on pg. 6) which represent different subgroups of interest in the dataset. The number of these regions varies from 20 to 715 in our datasets (see Tab. 3). One region corresponds to one instance of recourse verification problem (akin to a single data point in the pointwise setting). The computation times in Appendix E show avg. computation time over the regions (see line 1188). MIQCP solvers return the optimal solution for a given instance, so running each exp. multiple times would only show hardware variability in its comp. time.
>
> > The experiments also do not show results for changing the way the regions are calculated…
>
> We weren't quite sure what you meant here. Regions are given as input into our problem. We generate regions by looking at combinations of immutable features to represent subgroups (e.g., black female loan applicants). What other ways did you have in mind?
>
> > I am concerned that some of them are key into understanding the paper and should be either added to the main text or better referenced in the main text.
>
> Thanks for the suggestion! With the additional page in the camera ready version we’ll move some additional details on REP to the main paper.
>
> > Clarity, coherence and cohesion of the papers. The paper is unorganized (e.g. no subsections), information is sometimes repeated and key information is omitted. Giving the name when concepts are first mentioned in the introduction (RVP, ReVer...) would add a lot of clarity to the text .
>
> Thanks for the suggestion! In the revised draft we introduce the name of the key terms in the intro. Our draft uses paragraph subheadings (bold text) to structure the text - in a final draft we could replace those with subsections. If there are specific instances of missing information that you'd like us to fix please let us know!
>
> Thanks again, and please let us know if you have any additional questions.
>
> [1] Actionable Recourse in Linear Classification. Ustun et al.
>
> [2] Prediction without Preclusion: Recourse Verification with Reachable Sets. Kothari at al.
>
> [3] Explaining machine learning classifiers through diverse counterfactual explanations. Mothilal et al.

---

> > ### Comment · Reviewer_eTLW · 2025-04-04
> >
> > Thank you for addressing the concerns about the clarity and structure of the text and agreed to include all comments in the new draft version of the paper.
> >
> > About the experimental issues, the clarifications seem reasonable.
> >
> > Therefore, I will raise my score to 3.

---

### Official Review · Reviewer_6s3P · 2025-03-12

**Overall Recommendation:** 3

**Summary:**

This paper aims to identify regions where each point either allows recourse or does not allow recourse in the case of a linear model, given a fixed set of constraints.

In this specific case, the problem can be naturally formulated as a Mixed-Integer Quadratically Constrained Programming (MIQCP) problem. The authors also propose a solution for handling discrete variables more efficiently via Theorem 3.2 under some realistic assumptions.

**Claims And Evidence:**

All the claims presented in the paper are sound. However, my main concern is that from the title through the beginning of the paper (Section 2), it is not explicitly mentioned that the proposed solutions apply specifically to linear models.

**Essential References Not Discussed:**

N/A

**Experimental Designs Or Analyses:**

Yes, they are sound.

**Methods And Evaluation Criteria:**

Yes.

**Other Comments Or Suggestions:**

Please define more clearly the concept of a “fixed prediction” at the beginning of the paper—I struggled to fully grasp it until the formal definition was introduced. I recommend highlighting that the concept is tied to constraints or the action sets.

**Other Strengths And Weaknesses:**

N/A

**Questions For Authors:**

In the evaluation of the experiments, why restrict the feature space to fixed combinations? How many did you use in your experiments?

**Relation To Broader Scientific Literature:**

Existing solutions appear to have failed to solve the problem of interest, although their applicability extends to general models. The paper clearly demonstrates that the proposed solution is more effective for linear models.

**Theoretical Claims:**

Yes.

---

> ### Author Rebuttal · Authors · 2025-04-01
>
> Thank you for your time and feedback! We appreciate you highlighting that “existing solutions appear to have failed to solve the problem of interest”, our experiments are “sound”, and that our proposed approach is “more effective [than existing approaches] for linear models”.
>
> > However, my main concern is that from the title through the beginning of the paper (Section 2), it is not explicitly mentioned that the proposed solutions apply specifically to linear models.
>
> Thanks for flagging this! In the revised draft we’ve made sure to emphasize in the introduction and main contributions that this paper focuses specifically on linear models. We also want to highlight that our approach can be used for any MILP representable model but with weaker computational scaling (see discussion with Reviewer gp5s).
>
> > Please define more clearly the concept of a “fixed prediction” at the beginning of the paper—I struggled to fully grasp it until the formal definition was introduced. I recommend highlighting that the concept is tied to constraints or the action sets.
>
> Thanks for the suggestion! We’ve added a more clear description of fixed description (taken from Section 2) to the introduction to make the concept more clear.
>
> > In the evaluation of the experiments, why restrict the feature space to fixed combinations? How many did you use in your experiments?
>
> We restrict the feature space to fixed combinations of immutable features to represent sub-populations of interest (e.g., black female loan applicants). This also allows us to evaluate our approach on a number of different regions to provide more comprehensive experimental results (i.e., instead of one region per dataset). In our experiments we use between 20-715 regions (represented by $|\Psi|$ in our results table).
>
> Thanks again for the feedback - please let us know if you have any additional questions or concerns!

---

### Official Review · Reviewer_gp5S · 2025-03-14

**Overall Recommendation:** 3

**Summary:**

The paper  introduces a novel approach to identifying fixed predictions in machine learning models by finding confined regions where all individuals receive fixed predictions. The authors propose a method called ReVer, which uses mixed-integer quadratically constrained programming (MIQCP) to certify recourse for out-of-sample data and provide interpretable descriptions of confined regions. The paper emphasizes the importance of model responsiveness, especially in high-stakes settings like lending and hiring, and highlights the limitations of existing point-wise verification methods.

**Claims And Evidence:**

The authors claim that existing methods fail to discover confined regions and that ReVer can successfully identify these regions. They provide evidence through a comprehensive empirical study across various applications, demonstrating that ReVer can certify recourse and identify confined regions quickly and effectively. The paper also claims that ReVer is robust to distribution shifts and can be used without available datasets, which is supported by case studies and experiments.The authors claim that existing methods fail to discover confined regions and that ReVer can successfully identify these regions. They provide evidence through a comprehensive empirical study across various applications, demonstrating that ReVer can certify recourse and identify confined regions quickly and effectively. The paper also claims that ReVer is robust to distribution shifts and can be used without available datasets, which is supported by case studies and experiments.

**Essential References Not Discussed:**

NA

**Experimental Designs Or Analyses:**

The authors conduct experiments on three real-world datasets, evaluating ReVer's performance in terms of its ability to certify responsiveness and identify confined regions. They compare ReVer to point-wise baselines and analyze the results in terms of blindspots, loopholes, and computation time. The experiments demonstrate ReVer's effectiveness and efficiency in identifying confined regions and certifying recourse.

**Methods And Evaluation Criteria:**

The authors use MIQCP to develop ReVer, which identifies confined regions by verifying recourse over an entire region of the feature space. The evaluation criteria include the ability to certify responsiveness, identify confined regions, and run efficiently on real-world datasets. The authors compare ReVer to point-wise verification methods and evaluate its performance on datasets from consumer finance, content moderation, and criminal justice.

**Other Comments Or Suggestions:**

The authors could explore extending ReVer to non-linear models or other types of classifiers.
Providing more intuitive examples or visualizations of confined regions could help readers better understand the concept.

**Other Strengths And Weaknesses:**

+
The paper addresses a significant gap in the literature by focusing on confined regions rather than individual data points.
ReVer is shown to be effective and efficient in identifying confined regions and certifying recourse.
The method is robust to distribution shifts and can be used without available datasets, making it applicable in various settings.

-
The method is currently limited to linear classifiers, which may restrict its applicability to more complex models.
The paper could provide more detailed explanations of the mathematical formulations and assumptions.

**Questions For Authors:**

How do you plan to extend ReVer to handle non-linear models or more complex classifiers?
Can you provide more insights into the computational complexity of ReVer compared to other methods?
How does ReVer handle cases where the actionability constraints are not well-defined or are subject to change?

**Relation To Broader Scientific Literature:**

NA

**Theoretical Claims:**

The paper presents theoretical claims about the ability of ReVer to certify recourse over entire regions and provide guarantees for out-of-sample data. The authors also discuss the theoretical underpinnings of their approach, including the use of Farkas' lemma to certify infeasibility and the conditions under which their method can relax discrete variables.

---

> ### Author Rebuttal · Authors · 2025-04-01
>
> Thank you for your time and feedback! We were excited to see you recognized that our paper “addresses a significant gap in the literature”, “highlights the limitations of existing approaches”, and validates our claims via “a comprehensive empirical study across various applications”.
>
> We appreciate your comments and concerns, which have helped us to refine our paper. We would like to address them below:
> > The paper could provide more detailed explanations of the mathematical formulations and assumptions.
> > Providing more intuitive examples or visualizations of confined regions could help... readers better understand the topic..
> We agree! We plan on using the additional page for a camera ready version to expand on the assumptions for theorem 3.1, and move some of the discussion of the REP (currently in the appendix) to the main body of the paper. If there are any specific formulations or assumptions that were too vague, or additional visualizations beyond Figure 1 that you think would be helpful please let us know!
>
>
> > How do you plan to extend ReVer to handle non-linear models or more complex classifiers?
>
> Thanks for asking! Even though we focus on linear models, ReVer can work with **any MILP representable ML model** which includes decision trees, random forests, and even feed forward Neural Networks with ReLu activations (see [2] for more discussion). To adapt our framework, we just need to replace the linear classifier constraint (3a in the appendix, pg. 11) with constraints for the non-linear classifier. As an example, consider a one neuron neural network with weights beta, input x, and output v. We can model this as a MILP as follows (and replicate this for multiple neurons to embed an entire network):
> $$V \geq \beta^{\top} x $$
> $$V \leq  \beta^{\top} x - M(1 - z) $$
> $$V \leq Mz $$
> $$V \geq 0 $$
> $$z \in {0,1} $$
>
> In practice, modelling more complex ML models requires a lot of binary variables (e.g., z in the above example) which translates to a large number of continuous restrictions which could slow down the solve times of the MIQCP. If solving the MIQCP with all continuous restrictions is too demanding, we can always solve the linear relaxation with weaker guarantees: any confined box that ReVer returns is indeed confined, but ReVer cannot globally certify that a region is responsive (these nuances are discussed in Appendix D). To get global guarantees we would likely need to add some heavier duty optimization machinery to help tackle the large number of continuous restrictions (e.g., column and constraint generation). Our framework can also be used with local linear approximations of a more complex model (e.g., LIME).
>
> > Can you provide more insights into the computational complexity of ReVer compared to other methods?
>
> Verifying recourse for a single data point is itself an NP-hard problem. Similarly, the recourse verification over regions problem is NP-hard so we can’t expect any method with formal guarantees to run in polynomial time. In practice, MIQCPs are a bit harder to solve than MILPs (in our experiments solving one instance of ReVer takes about as long as solving 5 instances of pointwise verification). However, ReVer doesn’t scale with the number of data points to test, so it quickly outperforms pointwise verification in settings where we need to test more than a handful of data points. Detailed time comparison for all results are included in the supplementary materials.
>
> > How does ReVer handle cases where the actionability constraints are not well-defined or are subject to change?
>
> Great question! ReVer, like most of the existing literature on algorithmic recourse, assumes we have access to accurate and well-defined actionability constraints. In our experiments we focus on inherent actionability constraints that are intrinsic to the features (e.g., Age can only increase), which are by definition well defined and unlikely to change. In settings where the actionability constraints are not well-defined, ReVer could be combined with an iterative system to elicit actionability constraints (e.g., adapt [1]). Another big benefit of our MIQCP framework is that its flexible enough to incorporate different notions of robustness to address actionability constraints/classifiers that may chance  (e.g., instead of requiring $w^T (x + a) \geq 0$, we could replace 0 with a positive number to promote robustness). We’ll include this discussion in the revised draft.
>
> Thanks again and please let us know if you have additional questions!
>
> [1] Holy grail 2.0: From natural language to constraint models. Tsouros et al., arXiv (2023)
>
> [2] Mixed-integer optimization with constraint learning. Maragno et al., Operations Research (2023)

---

### Decision · Program_Chairs · 2025-05-01

**Decision:**

Accept (poster)

**Comment:**

This paper provides a new optimization method to certifiably identify regions where all individuals receive the same prediction from the model, where existing methods operate point-wise. The reviewers all found this a valuable contribution to the literature on recourse. The main weakness is that the optimization framework relies on being able to embed the predictive into a MILP. The existing paper focuses on linear models and it essential that the authors emphasize this heavily in the abstract/introduction. The authors point out that it is possible to embed various nonlinear kinds of models into a MILP but that the resulting problem may become computationally intractable to solve (although weaker guarantees can be obtained from the LP relaxation). In my mind, the paper would be a clear accept with a demonstration that some meaningful results could be obtained in the nonlinear case. As it is, the paper still makes a solid contribution as long as the important caveat of focusing on linear models is acknowledged more explicitly.